# ODGEN: Domain-specific Object Detection Data Generation with Diffusion Models

**Jingyuan Zhu**[1] **Shiyu Li**[2] **Yuxuan Liu**[2] **Jian Yuan**[1] **Ping Huang**[2]
**Jiulong Shan**[2] **Huimin Ma**[3]*

[1]Tsinghua University, China    [2]Apple    [3]University of Science and Technology Beijing

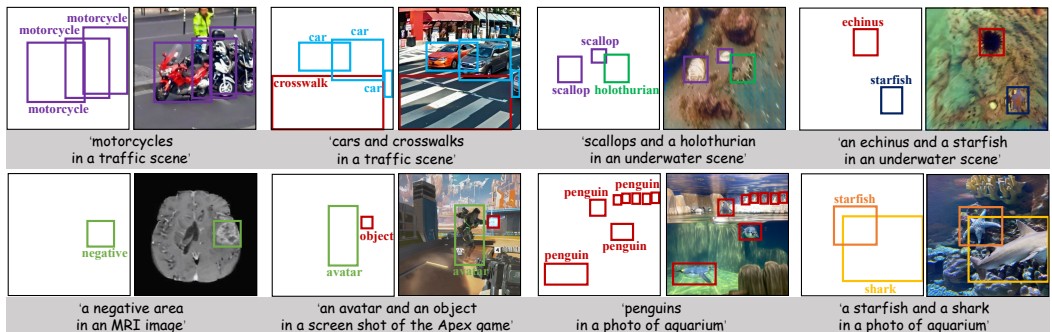

Figure 1: The proposed ODGEN enables controllable image generation from bounding boxes and text prompts. It can generate high-quality data for complex scenes, encompassing multiple categories, dense objects, and occlusions, which can be used to enrich the training data for object detection.

## Abstract

Modern diffusion-based image generative models have made significant progress and become promising to enrich training data for the object detection task. However, the generation quality and the controllability for complex scenes containing multi-class objects and dense objects with occlusions remain limited. This paper presents ODGEN, a novel method to generate high-quality images conditioned on bounding boxes, thereby facilitating data synthesis for object detection. Given a domain-specific object detection dataset, we first fine-tune a pre-trained diffusion model on both cropped foreground objects and entire images to fit target distributions. Then we propose to control the diffusion model using synthesized visual prompts with spatial constraints and object-wise textual descriptions. ODGEN exhibits robustness in handling complex scenes and specific domains. Further, we design a dataset synthesis pipeline to evaluate ODGEN on 7 domain-specific benchmarks to demonstrate its effectiveness. Adding training data generated by ODGEN improves up to 25.3% mAP@.50:.95 with object detectors like YOLOv5 and YOLOv7, outperforming prior controllable generative methods. In addition, we design an evaluation protocol based on COCO-2014 to validate ODGEN in general domains and observe an advantage up to 5.6% in mAP@.50:.95 against existing methods.

## 1 Introduction

Object detection is one of the most widely used computer vision tasks in real-world applications. However, data scarcity poses a significant challenge to the performance of state-of-the-art models

---

*H. Ma is the corresponding author.

38th Conference on Neural Information Processing Systems (NeurIPS 2024).

like YOLOv5 [25] and YOLOv7 [54]. With the progress of generative diffusion models, such as DALL-E [13], and Stable Diffision [44], which can generate high-quality images from text prompts, recent works have started to explore the potential of synthesizing images for perceptual model training. Furthermore, methods like GLIGEN [33], ReCo [61], and ControlNet [62] provide various ways to control the contents of synthetic images. Concretely, extra visual or textual conditions are introduced as guidance for diffusion models to generate objects under certain spatial constraints. Therefore, it becomes feasible to generate images together with instance-level or pixel-level annotations.

Nevertheless, it remains challenging to generate an effective supplementary training set for real-world object detection applications. Firstly, large-scale pre-trained diffusion models are usually trained on web crawl datasets such as LAION [48], whose distributions may be quite distinct from specialist domains. The domain gap could undermine the fidelity of generated images, especially the detailed textures and styles of the objects, e.g., the tumor in medical images or the avatar in a video game. Secondly, the text prompts for object detection scenes may contain multiple categories of subjects. It could cause the "concept bleeding" [12, 40, 68] problem for modern diffusion models, which is defined as the unintended merging or overlapping of distinct visual elements in an image. As a result, the synthetic images can be inconsistent with pseudo labels used as generative conditions. Thirdly, overlapping objects, which are common in object detection datasets, are likely to be merged as one single object by existing controllable generative methods. Lastly, some objects may be neglected by the diffusion model so that no foreground is generated at the conditioned location. All of the limitations can hinder the downstream object detection performance.

To address these challenges, we propose ODGEN, a novel method to generate synthetic images with a fine-tuned diffusion model and object-wise conditioning modules that control the categories and locations of generated objects. We first fine-tune the diffusion model on a given domain-specific dataset, with not only entire images but also cropped foreground patches, to improve the synthesis quality of both background scene and foreground objects. Secondly, we design a new text prompt embedding process. We propose to encode the class name of each object separately by the frozen CLIP [41] text encoder, to avoid mutual interference between different concepts. Then we stack the embeddings and encode them with a newly introduced trainable text embedding encoder. Thirdly, we propose to use synthetic patches of foreground objects as the visual condition. Each patch is resized and pasted on an empty canvas according to bounding box annotations, which deliver both conceptual and spatial information without interference from overlap between objects. In addition, we train foreground/background discriminators to check whether the region of each pseudo label contains a synthetic object. If no object can be found, the pseudo label will be filtered.

We evaluate the effectiveness of our ODGEN in specific domains on 7 subsets of the Roboflow-100 benchmark [6]. Extensive experimental results show that adding our synthetic data improves up to 25.3% mAP@.50:.95 on YOLO detectors, outperforming prior controllable generative methods. Furthermore, we validate ODGEN in general domains with an evaluation protocol designed based on COCO-2014 [37] and gain an advantage up to 5.6% in mAP@.50:.95 than prior methods.

The main contributions of this work can be summarized as follows:

- We propose to fine-tune pre-trained diffusion models with both cropped foreground patches and entire images to generate high-quality domain-specific target objects and background scenes.

- We design a novel strategy to control diffusion models with object-wise text prompts and synthetic visual conditions, improving their capability of generating and controlling complex scenes.

- We conduct extensive experiments to demonstrate that our synthetic data effectively improves the performance of object detectors. Our method outperforms prior works on fidelity and trainability in both specific and general domains.

## 2 Related Works

**Diffusion models** [7, 20, 29, 38, 49, 50] have made great progress in recent years and become mainstream for image generation, showing higher fidelity and training stability than prior generative models like GANs [4, 16, 26, 27, 28] and VAEs [30, 43, 52]. Latent diffusion models (LDMs) [44] perform the denoising process in the compressed latent space and achieve flexible generators conditioned on inputs like text and semantic maps. LDMs significantly improve computational efficiency and enable training on internet-scale datasets [48]. Modern large-scale text-to-image diffusion models

including eDiff-I [2], DALL·E [42], Imagen [46], and Stable Diffusion [40, 44] have demonstrated unparalleled capabilities to produce diverse samples with high fidelity given unfettered text prompts.

**Layout-to-image generation** synthesizes images conditioned on graphical inputs of layouts. Pioneering works [22, 35] based on GANs [16] and transformers [53] successfully generate images consistent with given layouts. LayoutDiffusion [66] fuses layout and category embeddings and builds object-aware cross-attention for local conditioning with traditional diffusion models. LayoutDiffuse [24] employs layout attention modules for bounding boxes based on LDMs. MultiDiffusion [3] and BoxDiff [58] develop training-free frameworks to produce samples with spatial constraints. GLIGEN [33] inserts trainable gated self-attention layers to pre-trained LDMs to fit specific tasks and is hard to generalize to scenes uncommon for pre-trained models. ReCo [61] and GeoDiffusion [5] extend LDMs with position tokens added to text prompts and fine-tunes both text encoders and diffusion models to realize region-aware controls. They need abundant data to build the capability of encoding the layout information in text prompts. ControlNet [62] reuses the encoders of LDMs as backbones to learn diverse controls. However, it still struggles to deal with some complex cases. MIGC [67] decomposes multi-instance synthesis to single-instance subtasks utilizing additional attention modules. InstanceDiffusion [55] adds precise instance-level controls, including boxes, masks, and scribbles for text-to-image generation. Given annotations of bounding boxes, this paper introduces a novel approach applicable to both specific and general domains to synthesize high-fidelity complex scenes consistent with annotations for the object detection task.

**Dataset synthesis for training object detection models.** Copy-paste [9] is a simple but effective data augmentation method for detectors. Simple Copy-Paste [14] achieves improvements with a simple random placement strategy for objects. Following works [13, 65] generate foreground objects and then paste them on background images. However, generated images by these methods usually have obvious artifacts on the boundary of pasted regions. DatasetGAN [64] takes an early step to generate labeled images automatically. Diffusion models have been used to synthesize training data and benefit downstream tasks including object detection [5, 10, 11, 63], image classification [1, 8, 18, 34, 47, 51], and semantic segmentation [15, 23, 31, 36, 39, 56, 57, 59, 60]. Modern approaches for image-label pairs generation can be roughly divided into two categories. One group of works [11, 15, 31, 36, 56, 59, 63] first generate images and then apply a pre-trained perception model to generate pseudo labels on the synthetic images. The other group [5, 23, 39] uses the same strategy as our approach to synthesize images under the guidance of annotations as input conditions. The second group also overlaps with layout-to-image approaches [24, 33, 55, 61, 62, 66, 67]. Our approach should be assigned to the second group and is designed to address challenging cases like multi-class objects, occlusions between objects, and specific domains.

# 3   Method

This section presents ODGEN, a novel approach to generate high-quality images conditioned on bounding box labels for specific domains. Firstly, we propose a new method to fine-tune the diffusion model in Sec. 3.1. Secondly, we design an object-wise conditioning method in Sec. 3.2. Finally, we introduce a generation pipeline to build synthetic datasets for detector training enhancement in Sec. 3.3.

## 3.1   Domain-specific Diffusion Model Fine-tuning

Modern Stable Diffusion [44] models are trained on the LAION-5B [48] dataset. Therefore, the textual prompts and generated images usually follow similar distributions to the web crawl data. However, real-world object detection datasets could come from very specific domains. We fine-tune the UNet of the Stable Diffusion to fit the distribution of an arbitrary object detection dataset. For training images, we use not only the entire images from the dataset but also the crops of foreground objects. In particular, we crop foreground objects and resize them to $512 \times 512$. In terms of the text input, as shown in Fig. 2 (a), we use templated condition "a $<scene>$" named $c_s$ for entire images and "a $<classname>$" named $c_o$ for cropped objects. If the names are terminologies, following the subject-driven approach in DreamBooth [45], we can employ identifiers like "[V]" to represent the target objects and scenes, improving the robustness to textual ambiguity under the domain-specific context. $x_s^t$ and $x_o^t$ represent the input of scenes $x_s$ and objects $x_t$ added to noises at time step $t$. $\epsilon_s$ and $\epsilon_o$ represent the added noises following normal Gaussian distributions. The pre-trained Stable

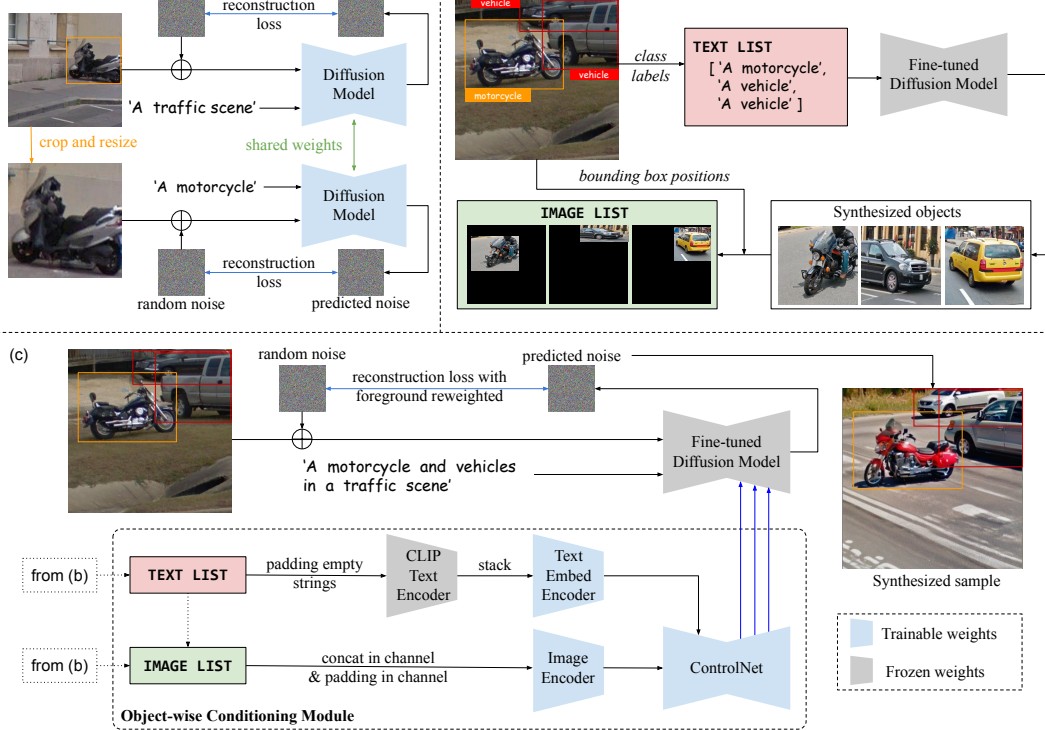

Figure 2: ODGEN training pipeline: (a) A pre-trained diffusion model is fine-tuned on a detection dataset with both entire images and cropped foreground patches. (b) A text list is built based on class labels. The fine-tuned diffusion model in stage (a) is used to generate a synthetic object image for each text. Generated object images are resized and pasted on empty canvases per box positions, constituting an image list. (c) The image list is concatenated in the channel dimension and encoded as conditions for ControlNet. The text list is encoded by the CLIP text encoder, stacked, and encoded again by the text embedding encoder as inputs for ControlNet.

Diffusion parameterized by $\theta$ is fine-tuned with the reconstruction loss as:

$$\begin{aligned}
\mathcal{L}_{rec} =& \mathbb{E}_{x_o,t,\epsilon_o \sim \mathcal{N}(0,1)} \left[ ||\epsilon_o - \epsilon_\theta(x_o^t, t, \tau(c_o))||^2 \right] \\
& + \lambda \mathbb{E}_{x_s,t,\epsilon_s \sim \mathcal{N}(0,1)} \left[ ||\epsilon_s - \epsilon_\theta(x_s^t, t, \tau(c_s))||^2 \right]
\end{aligned} \tag{1}$$

where $\lambda$ controls the relative weight for the reconstruction loss of scene images. $\tau$ is the frozen CLIP text encoder. Our approach guides the fine-tuned model to capture more details of foreground objects and maintain its capability of synthesizing the complete scenes.

## 3.2 Object-wise Conditioning

ControlNet [62] can perform controllable synthesis with visual conditions. However, the control can be challenging with an increasing object number and category number due to the "concept bleeding" phenomenon. Stronger conditions are needed to ensure high-fidelity generation. To this end, we propose a novel object-wise conditioning strategy with ControlNet.

**Text list encoding.** As shown in Fig. 2 (b) and (c), given a set of object class names and their bounding boxes, we first build a text list consisting of each object with the fixed template "a *<classname>*". Then the text list is padded with empty texts to a fixed length $N$ and converted to a list of embeddings by a pre-trained CLIP text tokenizer and encoder. The embeddings are stacked and encoded by a 4-layer convolutional text embedding encoder, and used as the textual condition for ControlNet. The text encoding of native ControlNet compresses the information in global text prompts altogether, resulting in mutual interference between distinct categories. To alleviate this "concept bleeding" problem of multiple categories, our two-step text encoding enables the ControlNet to capture the information of each object with separate encoding.

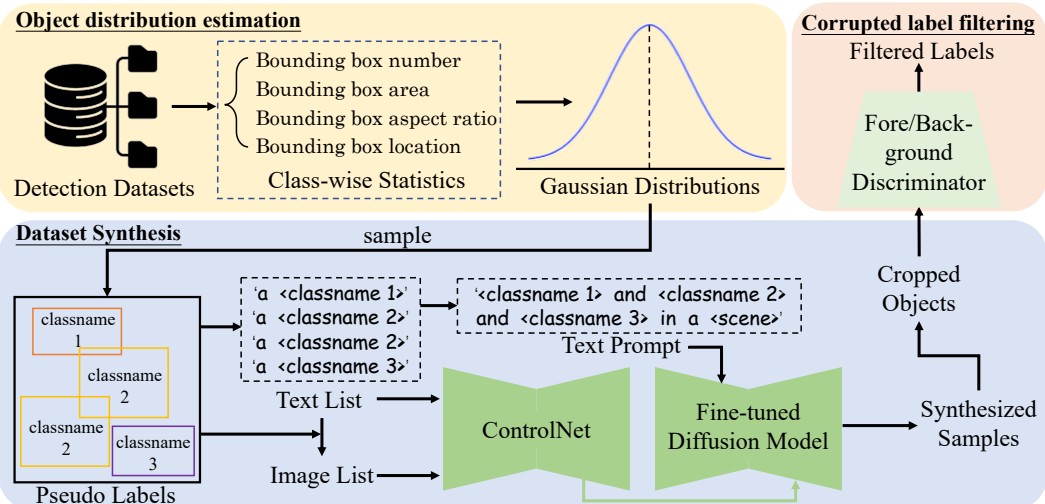

Figure 3: Pipeline for object detection dataset synthesis. **Yellow block**: estimate Gaussian distributions for the bounding box number, area, aspect ratio, and location based on the training set. **Blue block**: sample pseudo labels from the Gaussian distributions and generate conditions including text and image lists to synthesize novel images. **Pink block**: train a classifier with foreground and background patches randomly cropped from the training set and use it to filter pseudo labels that failed to be synthesized. Finally, the filtered labels and synthetic images compose datasets.

**Image list encoding.** As shown in Fig. 2 (b) and (c), for each item in the text list, we use the fine-tuned diffusion model to generate images for each foreground object. We then resize each generated image and paste it on an empty canvas based on the corresponding bounding box. The set of pasted images is denoted as an image list, which contains both conceptual and spatial information of the objects. The image list is concatenated and zero-padded to $N$ in the channel dimension and then encoded to the size of latent space by an image encoder. Applying an image list rather than pasting all objects on a single image can effectively avoid the influence of object occlusions.

**Optimization.** With a pair of image and object detection annotation, we generate the text list $c_{tl}$ and image list $c_{il}$ as introduced in this section. The input global text prompt $c_t$ of the diffusion model is composed of the object class names and a scene name, which is usually related to the dataset name and fixed for each dataset. The ControlNet, along with the integrated encoders, are trained with the reconstruction loss. In addition, the weights of foreground regions are enhanced to produce more realistic foreground objects in complex scenes. The overall loss function is formulated as:

$$\mathcal{L}_{recon} = \mathbb{E}_{x,t,c_t,c_{tl},c_{il},\epsilon\sim\mathcal{N}(0,1)} \left[||\epsilon - \epsilon_\theta(x_t, t, c_t, c_{tl}, c_{il})||^2\right]$$
$$\mathcal{L}_{control} = \mathcal{L}_{recon} + \gamma\mathcal{L}_{recon} \odot \mathcal{M} \tag{2}$$

where $\mathcal{M}$ represents a binary mask with 1 on foreground pixels 0 on background pixels. $\odot$ represents element-wise multiplication, and $\gamma$ controls the foreground re-weighting.

### 3.3 Dataset Synthesis Pipeline for Object Detection

Our dataset synthesis pipeline is summarized in Fig. 3. We generate random bounding boxes and classes as pseudo labels based on the distributions of the training set. Then, the pseudo labels are converted to triplets: <image list, text list, global text prompt>. ODGEN uses the triplets as inputs to synthesize images with the fine-tuned diffusion model. In addition, we filter out the pseudo labels in which areas the foreground objects fail to be generated.

**Object distribution estimation.** To simulate the training set distribution, we compute the mean and variance of bounding box attributes and build normal distributions. In particular, given a dataset with $K$ categories, we calculate the class-wise average occurrence number per image $\boldsymbol{\mu} = (\mu_1, \mu_2, \ldots, \mu_K)$ and the covariance matrix $\boldsymbol{\Sigma}$ to build a multi-variable joint normal distribution $\mathcal{N}(\boldsymbol{\mu}, \boldsymbol{\Sigma})$. In addition, for each category $k$, we build normal distributions $\mathcal{N}(\mu_x, \sigma_x^2)$ and $\mathcal{N}(\mu_y, \sigma_y^2)$

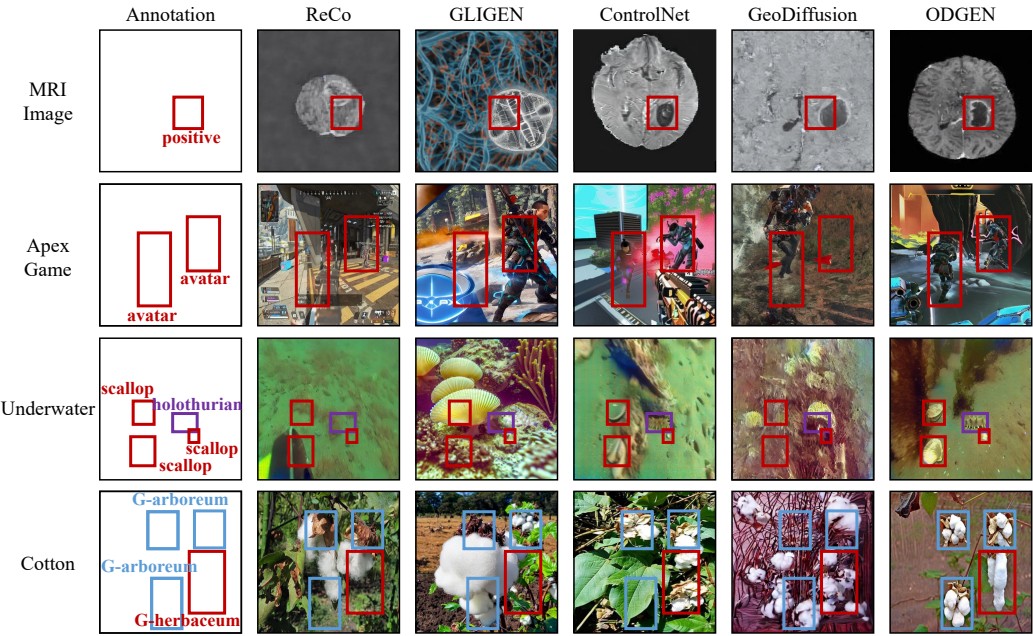

Figure 4: Comparison between ODGEN and other methods under the same condition shown in the first column. ODGEN can be generalized to specific domains and enables accurate layout control.

for the top-left coordinate of the bounding boxes, $\mathcal{N}(\mu_{area}, \sigma_{area}^2)$ for box areas, and distributions $\mathcal{N}(\mu_{ratio}, \sigma_{ratio}^2)$ for aspect ratios.

**Image synthesis.** We first sample the object number per category from the $K$-dimensional joint normal distributions. Then for each object, we sample its bounding box location and size from the corresponding estimated normal distributions. The sampled values are used to generate text lists, image lists, and global text prompts. Finally, we use ODGEN to generate images with the triplets.

**Corrupted label filtering.** There is a certain possibility that some objects fail to be generated in synthetic images. While utilizing image lists alleviates the concern to some extent, the diffusion model may still neglect some objects in complex scenes containing dense or overlapping objects. As a result, the synthesized pixels could not match the pseudo labels and will undermine downstream training performance. We fine-tune a ResNet50 [17] with foreground and background patches cropped from the training set to classify whether the patch contains an object. The discriminator checks whether objects are successfully synthesized in the regions of pseudo labels, rejects nonexistent ones, and further improves the accuracy of synthetic datasets.

## 4   Experiments

We conduct extensive experiments to demonstrate the effectiveness of the proposed ODGEN in both specific and general domains. FID [19] is used as the metric of fidelity. Mean average precisions (mAP) of YOLOv5s [25] and YOLOv7 [54] trained with synthetic data are used to evaluate the trainability, which concerns the usefulness of synthetic data for detector training. Our approach is implemented with Stable Diffusion v2.1 and compared with prior controllable generation methods based on Stable Diffusion, including ReCo [61], GLIGEN [33], ControlNet [62], GeoDiffusion [5], InstanceDiffusion [55], and MIGC [67]. Native ControlNet doesn't support bounding box conditions. Therefore, we convert boxes to a mask $C$ sized $H \times W \times K$, where $H$ and $W$ are the image height and width, and $K$ is the class number. If one pixel $(i, j)$ is in $n$ boxes of class $k$, the $C[i, j, k] = n$. YOLO models are trained with the same recipe as Roboflow-100 [6] for 100 epochs to ensure convergence. The length of text and image lists $N$ used in our ODGEN is set to the maximum object number per image in the training set.

Table 1: FID (↓) scores computed over 5000 images synthesized by each approach on RF7 datasets. ODGEN achieves better results than the other on all 7 domain-specific datasets.

| Datasets | ReCo | GLIGEN | ControlNet | GeoDiffusion | ODGEN |
|---|---|---|---|---|---|
| Apex Game | 88.69 | 125.27 | 97.32 | 120.61 | **58.21** |
| Robomaster | 70.12 | 167.44 | 134.92 | 76.81 | **57.37** |
| MRI Image | 202.36 | 270.52 | 212.45 | 341.74 | **93.82** |
| Cotton | 108.55 | 89.85 | 196.87 | 203.02 | **85.17** |
| Road Traffic | 80.18 | 98.83 | 162.27 | 68.11 | **63.52** |
| Aquarium | 122.71 | 98.38 | 146.26 | 162.19 | **83.07** |
| Underwater | 73.29 | 147.33 | 126.58 | 125.32 | **70.20** |

Table 2: mAP@.50:.95 (↑) of YOLOv5s / YOLOv7 on RF7. Baseline models are trained with 200 real images only, whereas the other models are trained with 200 real + 5000 synthetic images from various methods. ODGEN leads to the biggest improvement on all 7 domain-specific datasets.

| | Baseline | ReCo | GLIGEN | ControlNet | GeoDiffusion | ODGEN |
|---|---|---|---|---|---|---|
| real + synth # | 200 + 0 | 200 + 5000 | 200 + 5000 | 200 + 5000 | 200 + 5000 | 200 + 5000 |
| Apex Game | 38.3 / 47.2 | 25.0 / 31.5 | 24.8 / 32.5 | 33.8 / 42.7 | 29.2 / 35.8 | **39.9 / 52.6** |
| Robomaster | 27.2 / 26.5 | 18.2 / 27.9 | 19.1 / 25.0 | 24.4 / 32.9 | 18.2 / 22.6 | **39.6 / 34.7** |
| MRI Image | 37.6 / 27.4 | 42.7 / 38.3 | 32.3 / 25.9 | 44.7 / 37.2 | 42.0 / 38.9 | **46.1 / 41.5** |
| Cotton | 16.7 / 20.5 | 29.3/ 37.5 | 28.0 / 39.0 | 22.6 / 35.1 | 30.2 / 36.0 | **42.0 / 43.2** |
| Road Traffic | 35.3 / 41.0 | 22.8 / 29.3 | 22.2 / 29.5 | 22.1 / 30.5 | 17.2 / 29.4 | **39.2 / 43.8** |
| Aquarium | 30.0 / 29.6 | 23.8 / 34.3 | 24.1 / 32.2 | 18.2 / 25.6 | 21.6 / 30.9 | **32.2 / 38.5** |
| Underwater | 16.7 / 19.4 | 13.7 / 15.8 | 14.9 / 18.5 | 15.5 / 17.8 | 13.8 / 17.2 | **19.2 / 22.0** |

## 4.1 Specific Domains

Roboflow-100 [6] is used for the evaluation in specific domains. It consists of 100 object detection datasets of various domains. We select 7 representative datasets (RF7) covering video games, medical imaging, and underwater scenes. To simulate data scarcity, we only sample 200 images as the training set for each dataset. The whole training process including the fine-tuning on both cropped objects and entire images and the training of the object-wise conditioning module, only depends on the 200 images. $\lambda$ in Eq. (1) and $\gamma$ in Eq. (2) are set as 1 and 25. We first fine-tune the diffusion model according to Fig. 2 (a) for 3k iterations. Then we train the object-wise conditioning modules on a V100 GPU with batch size 4 for 200 epochs, the same as the other methods to be compared.

### 4.1.1 Fidelity

For each dataset in RF7, we compute the FID scores of 5000 synthetic images against real images, respectively. As shown in Tab. 1, our approach outperforms all the other methods. We visualize the generation quality in Fig. 4. GLIGEN only updates gated self-attention layers inserted to Stable Diffusion, making it hard to be generalized to specific domains like MRI images and the Apex game. ReCo and GeoDiffusion integrate encoded bounding box information into text tokens, which require more data for diffusion model and text encoder fine-tuning. With only 200 images, they fail to generate objects in the given box regions. ControlNet, integrating the bounding box information into the visual condition, presents more consistent results with the annotation. However, it may still miss some objects or generate wrong categories in complex scenes. ODGEN achieves superior performance on both visual effects and consistency with annotations.

### 4.1.2 Trainability

To explore the effectiveness of synthetic data for detector training, we train YOLO models pre-trained on COCO with different data and compare their mAP@.50:.95 scores in Tab. 2. In the baseline setting, we only use the 200 real images for training. For the other settings, we use a combination of 200 real and 5000 synthetic images. Our approach gains improvement over the baseline and outperforms all the other methods.

In addition, we add experiments with larger-scale datasets with 1000 images sampled from the Apex Game and the Underwater datasets. The training setups are kept the same as the training on 200 images. We conduct experiments on ODGEN, ReCo, and GeoDiffusion. ReCo and GeoDiffusion

Table 3: mAP@.50 / mAP@.50:.95 (↑) results of ODGEN trained on larger-scale datasets of 1000 real images. The top 3 rows show results of YOLOv5s and the bottom 3 rows show results of YOLOv7. Baseline models are trained with 1000 real images only, whereas the other models are trained with 1000 real + 5000 / 10000 synthetic images from various methods. ODGEN leads to more significant improvement than other methods.

| Datasets | Apex Game | Apex Game | Apex Game | Underwater | Underwater | Underwater |
|---|---|---|---|---|---|---|
| real + synth # | 1000 + 0 | 1000 + 5000 | 1000 + 10000 | 1000 + 0 | 1000 + 5000 | 1000 + 10000 |
| ReCo | 83.2 / 53.5 | 78.7 / 46.9 | 82.0 / 46.9 | 55.6 / 29.2 | 55.1 / 28.4 | 55.9 / 29.1 |
| GeoDiffusion | 83.2 / 53.5 | 80.0 / 47.2 | 82.5 / 47.5 | 55.6 / 29.2 | 54.2 / 27.9 | 54.3 / 28.0 |
| ODGEN | 83.2 / 53.5 | **83.3 / 53.5** | **83.6 / 53.6** | 55.6 / 29.2 | **59.6 / 32.5** | **56.3 / 29.8** |
| ReCo | 83.8 / 55.0 | 80.5 / 50.7 | 79.2 / 49.9 | 54.6 / 28.3 | 56.5 / 28.7 | 56.4 / 30.1 |
| GeoDiffusion | 83.8 / 55.0 | 81.2 / 51.0 | 81.0 / 50.5 | 54.6 / 28.3 | 57.0 / 28.9 | 55.8 / 28.9 |
| ODGEN | 83.8 / 55.0 | **84.4 / 55.2** | **84.0 / 55.0** | 54.6 / 28.3 | **58.2 / 29.8** | **62.1 / 31.8** |

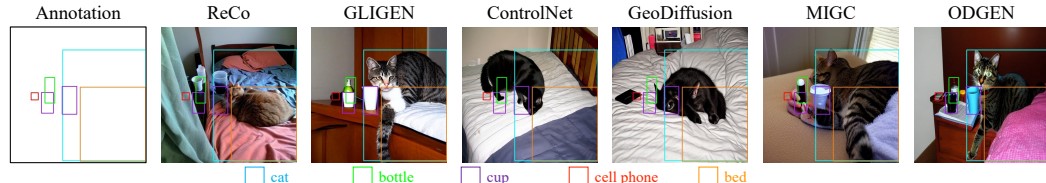

Figure 5: Visualized results comparison for models trained on COCO. ODGEN is better qualified for synthesizing complex scenes with multiple categories of objects and bounding box occlusions.

may benefit from larger-scale training datasets since they need to fine-tune more parameters in both the UNet in Stable Diffusion and the CLIP text encoder. GLIGEN struggles to adapt to new domains. ControlNet performs worse on layout control than ODGEN. Therefore, these two methods are not included in this part. The corrupted label filtering step is not used for any method. Results in Tab. 3 show that the baselines trained on real data only become stronger with larger-scale datasets while our ODGEN still benefits detectors with synthetic data and outperforms other methods.

Table 4: FID (↓) and mAP (↑) of YOLOv5s / YOLOv7 on COCO. FID is computed with 41k synthetic images. For mAP, YOLO models are trained from scratch on 10k synthetic images and validated on 31k real images. ODGEN outperforms all the other methods in terms of both fidelity and trainability.

| Metrics | ReCo | GLIGEN | ControlNet | Geo-Diffusion | MIGC | Instance-Diffusion | ODGEN |
|---|---|---|---|---|---|---|---|
| FID | 18.36 | 26.15 | 25.54 | 30.00 | 21.82 | 23.29 | **16.16** |
| mAP@.50 | 7.60 / 11.01 | 6.70 / 9.42 | 1.43 / 1.15 | 5.94 / 9.21 | 9.54 / 16.01 | 10.00 / 17.10 | **18.90 / 24.40** |
| mAP@.50:.95 | 3.82 / 5.29 | 3.56 / 4.60 | 0.52 / 0.38 | 2.37 / 4.44 | 4.67 / 8.65 | 5.42 / 10.20 | **9.70 / 14.20** |

## 4.2 General Domains

COCO-2014 [37] is used to evaluate our method in general domains. As the scenes in this dataset are almost covered by the pre-training data of Stable Diffusion, we skip the domain-specific fine-tuning stage and train the diffusion model together with the object-wise conditioning modules. We train our ODGEN on the COCO training set with batch size 32 for 60 epochs on ×8 V100 GPUs as well as GLIGEN, ControlNet, and GeoDiffusion. Officially released checkpoints of ReCo (trained for 100 epochs on COCO), MIGC (trained for 300 epochs on COCO), and InstanceDiffusion (trained on self-constructed datasets) are employed for comparison. $\gamma$ in Eq. (2) is set as 10 for our ODGEN.

### 4.2.1 Fidelity

We use the annotations of the COCO validation set as conditions to generate 41k images. The FID scores in Tab. 4 are computed against the COCO validation set. ODGEN achieves better FID results than the other methods. We provide a typical visualized example in Fig. 5. Given overlapping bounding boxes of multiple categories, ODGEN synthesizes all the objects of correct categories and accurate positions, outperforming the other methods. More samples are added in Appendix G.

Table 5: mAP@.50 (↑) and mAP@.50:.95 (↑) of YOLOv5s / YOLOv7 on the COCO dataset. Baseline models are trained with 80k images from the COCO training set, whereas the other models are trained with the same 80k real + 20k synthetic images. ODGEN outperforms baseline and the other methods.

| Metrics | Baseline | ReCo | GLIGEN | ControlNet |
|---|---|---|---|---|
| mAP@.50 | 51.5 / 64.5 | 50.8 / 64.3 | 50.9 / 64.2 | 50.1 / 64.3 |
| mAP@.50:.95 | 32.6 / 45.4 | 31.8 / 45.2 | 31.9 / 45.2 | 31.0 / 45.2 |
| Metrics | GeoDiffusion | MIGC | InstanceDiffusion | ODGEN |
| mAP@.50 | 51.2 / 64.4 | 51.5 / 64.6 | 51.5 / 64.6 | **52.1 / 65.0** |
| mAP@.50:.95 | 32.1 / 45.2 | 32.5 / 45.5 | 32.6 / 45.6 | **33.1 / 45.9** |

### 4.2.2 Trainability

We use 10k annotations randomly sampled from the COCO validation set to generate a synthetic dataset of 10k images. The YOLO models are trained on this synthetic dataset from scratch and evaluated on the other 31k images in the COCO validation set. As shown in Tab. 4, ODGEN achieves significant improvement over the other methods in terms of mAP.

We further conduct experiments by adding 20k synthetic images to the 80k training images. We train YOLO models from scratch on the COCO training set (80k images) as the baseline and on the same 80k real images + 20k synthetic images generated by different methods for comparison. We use the labels of 20k images from the COCO validation set as conditions to generate the synthetic set and use the other 21k real images for evaluation. The results are shown in Tab. 5. It shows that ODGEN improves the mAP@.50:95 by 0.5% and outperforms the other methods.

We observe that YOLO models trained on synthetic data only fall behind models trained on real data. We add experiments with different training and validation data combinations in Tab. 6. YOLO models trained on real images show better generalization ability and achieve close results when tested on real and synthetic data. YOLO models trained on synthetic data only get significantly better results when tested on synthetic data than when generalized to real data. It indicates that noticeable domain gaps still exist between real and synthetic data, which may be limited by the generation quality of modern Stable Diffusion models and are promising to be narrowed with future models.

Table 6: mAP@.50 (↑) and mAP@.50:.95 (↑) of YOLOv5s / YOLOv7 trained from scratch and validated on real or synthetic COCO validation set. 10k images are used for training and the other 31k images are used for validation. Real represents real images in the COCO validation set and synthetic represents images synthesized by our ODGEN using the same labels.

| Train | Validate | mAP@.50 (↑) | mAP@.50:95 (↑) | Train | Validate | mAP@.50 (↑) | mAP@.50:95 (↑) |
|---|---|---|---|---|---|---|---|
| Real | Real | 29.40 / 41.70 | 15.80 / 26.30 | Synthetic | Real | 18.90 / 24.40 | 9.70 / 14.20 |
| Real | Synthetic | 29.40 / 39.90 | 15.00 / 23.20 | Synthetic | Synthetic | 37.90 / 45.40 | 20.40 / 27.10 |

Table 7: Using the image list (IL) and the text list (TL) benefits FID and mAP (YOLOv5s / YOLOv7). They are especially helpful for the Road Traffic dataset which has more categories and occlusions.

| Datasets | IL | TL | FID(↓) | mAP@.50:.95 (↑) |
|---|---|---|---|---|
| MRI Image | × | × | 98.29 | 44.6 / 39.9 |
| | × | ✓ | 95.67 | 44.9 / 41.4 |
| | ✓ | × | 99.16 | **46.2** / 41.0 |
| | ✓ | ✓ | **93.82** | 46.1 / **41.5** |
| Road Traffic | × | × | 67.40 | 25.5 / 35.4 |
| | × | ✓ | 66.48 | 26.5 / 37.0 |
| | ✓ | × | 65.80 | 32.3 / 39.1 |
| | ✓ | ✓ | **63.52** | **39.2 / 43.8** |

Table 8: Proper $\gamma$ values in Eq. (2) benefit both FID and mAP (YOLOv5s / YOLOv7) of synthetic images, while overwhelming values lead to degeneration.

| Datasets | $\gamma$ | FID (↓) | mAP@.50:.95 (↑) |
|---|---|---|---|
| Aqua-rium | 0 | 83.94 | 30.8 / 34.5 |
| | 25 | **83.07** | **32.2 / 38.5** |
| | 50 | 87.00 | 32.1 / 38.0 |
| | 100 | 90.13 | 29.9 / 35.7 |
| Road Traffic | 0 | 66.65 | 36.9 / 39.5 |
| | 25 | **63.52** | **39.2 / 43.8** |
| | 50 | 66.46 | 36.6 / 38.9 |
| | 100 | 72.18 | 32.9 / 37.1 |

### 4.3 Ablation Study

**Image list and text list in object-wise conditioning modules.** ODGEN enables object-wise conditioning with an image list and a text list as shown in Fig. 2. We test the performance of

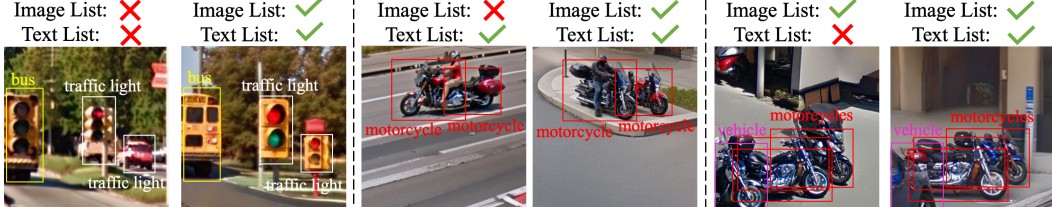

Figure 6: Visualized ablations. **Left**: using neither image nor text lists, a traffic light is generated in the position of a bus, and a car is generated in the position of a traffic light; **Middle**: not using image list, two occluded motorcycles are merged as one; **Right**: not using text list, a motorcycle is generated in the position of a vehicle. Using both image and text lists generates correct results.

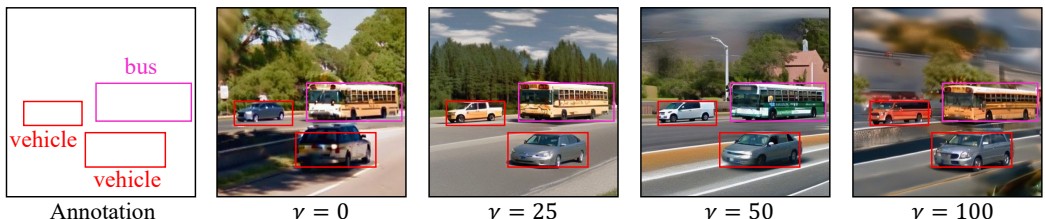

Figure 7: With increasing $\gamma$ value, our approach produces higher-quality foreground objects. However, overwhelming $\gamma$ leads to blurred background and degenerated fidelity.

using global text prompts in the placement of text lists or pasting all the foreground patches on the same empty canvas in the placement of image lists for the ablation study. Experiments are performed on the MRI and Road Traffic datasets in RF7. The MRI dataset only has two categories and almost no overlapping objects, whereas the road traffic dataset has 7 categories and many occluded objects. From the results in Tab. 7, it can be found that using image lists or text lists brings moderate improvement on MRI images and significant improvement on the more complex Road Traffic dataset. Visualization in Fig. 6 shows that the image list improves the fidelity under occlusion, and the text list mitigates the mutual interference of multiple objects.

**Foreground region enhancement.** We introduce a re-weighting term controlled by $\gamma$ for foreground objects in Eq. (2). Tab. 8 shows that appropriate $\gamma$ values can improve both the fidelity and the trainability since the foreground details are better generated. However, increasing the value will undermine the background quality, as visualized in Fig. 6.

**Corrupted label filtering.** Results in Tab. 9 show that the corrupted label filtering step helps improve the mAP@.50:95 of ODGEN by around 1-2%. The results of ODGEN without the corrupted label filtering still significantly outperform the baseline setting in Tab. 2.

Table 9: mAP (YOLOv5s / YOLOv7) of the corrupted label filtering for ODGEN.

| Datasets | Label filtering | mAP@.50:.95 (↑) |
|---|---|---|
| Cotton | ✓ | **42.0 / 43.2** |
| | ✗ | 40.5 / 42.1 |
| Robo-master | ✓ | **39.6 / 34.7** |
| | ✗ | 39.0 / 33.3 |
| Under-water | ✓ | **19.2 / 22.0** |
| | ✗ | 18.9 / 21.6 |

## 5 Conclusion

This paper introduces ODGEN, a novel approach aimed at generating domain-specific object detection datasets to enhance the performance of detection models. We propose to fine-tune the diffusion model on both cropped foreground objects and entire images. We design a mechanism to control the foreground objects with object-wise synthesized visual prompts and textual descriptions. Our work significantly enhances the performance of diffusion models in synthesizing complex scenes characterized by multiple categories of objects and bounding box occlusions. Extensive experiments demonstrate the superiority of ODGEN in terms of fidelity and trainability across both specific and general domains. We believe this work has taken an essential step toward more robust image synthesis and highlights the potential of benefiting the object detection task with synthetic data.

# Acknowledgements

We thank support from National Natural Science Foundation of China under Grant 62227801 and National Science and Technology Major Project under Grant 2022ZD0117902.

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

## A    Limitations

Despite the significant improvement achieved by this work, we find it remains challenging to achieve comparable training performance as real data with merely synthetic data. We consider the limitation to be mainly caused by two reasons. Firstly, the generation fidelity is still limited by modern diffusion models. For instance, VAEs in Stable Diffusion are not good enough to deal with complex text in images, constraining the performance on video game datasets. ODGEN is promising to achieve further improvements with more powerful diffusion models in the future. Secondly, this paper mainly focuses on designing generative models with given conditions. However, the other parts in the dataset synthesis pipeline are not fully optimized, which are interesting to be explored in future work.

## B    Broader Impact

We introduce a novel approach for object detection datasets synthesis, achieving improvement in fidelity and trainability for both specific and general domains. Our approach is more prone to biases caused by training data than typical AI generative models since it shows robustness with limited data of specific domains. Therefore, we recommend practitioners to apply abundant caution when dealing with sensitive applications to avoid problems of races, skin tones, or gender identities.

## C    Supplemental Ablations

**Detector training w/o real data.** In the main paper, we train the detectors with 200 real images and 5000 synthetic images for RF7 benchmarks. In this ablation study, we conduct experiments with the 5000 synthetic images only to explore the impact of the absence of real data for training detectors. As shown in Tab. 10, for easy datasets with fewer objects, like MRI image and Cotton, training with 5000 synthetic data only achieves better results than training with 200 real samples only. Increasing the dataset size can improve the model performance because the generation quality is good enough in these cases. Nevertheless, for complex scenes like Robomaster and Aquarium, training with synthetic data only fails to achieve better results. It may be caused by the limited generation quality of modern diffusion models. For instance, the VAE in Stable Diffusion is not good at synthesizing texts in images, which is common in the Robomaster dataset. Besides, FID results in Tab. 1 indicate that there still exist gaps between the distributions of real and synthetic images. The performance of our approach is promising to be further improved in the future with more powerful generative diffusion models. In addition, the current dataset synthesis pipeline is not fully optimized and cannot reproduce the distributions of foreground objects perfectly, which may also lead to worse performance of training on synthetic data only with the RF7 datasets.

Table 10: Ablations of using real data only and using synthetic data only during detector training. (Metric: YOLOv5s / YOLOv7)

| Metrics | mAP@.50 (↑) | | | mAP@.50:.95 (↑) | | |
|---|---|---|---|---|---|---|
| real + synth # | 200 + 0 | 0 + 5000 | 200 + 5000 | 200 + 0 | 0 + 5000 | 200 + 5000 |
| Robomaster | 50.3 / 48.8 | 35.9 /37.2 | 67.4 / 65.4 | 27.2 / 26.5 | 18.7 / 21.0 | 39.6 / 34.7 |
| MRI Image | 55.1 / 44.7 | 59.8 / 55.2 | 68.1 / 61.3 | 37.6 / 27.4 | 41.3 / 35.3 | 46.1 / 41.5 |
| Cotton | 29.4 / 32.0 | 43.1 / 43.2 | 63.9 / 55.0 | 16.7 / 20.5 | 29.5 / 28.1 | 42.0 / 43.2 |
| Aquarium | 53.8 / 53.0 | 45.2 / 41.1 | 62.6 / 66.5 | 30.0 / 29.6 | 23.8 / 20.9 | 32.2 / 38.5 |

**Number of synthesized samples.** We ablate the number of synthesized samples used for detector training and provide quantitative results in Tab. 11. With increasing amounts of synthesized samples, detectors achieve higher performance and get very close results between 5000 and 10000 synthetic samples.

**Category isolation for datasets synthesis.** As illustrated in the main paper, modern diffusion models have the limitation of "concept bleeding" when multiple categories are involved in the generation of one image. An alternative method could be only generating objects of the same category in one image. We select 3 multi-class datasets and compare this approach against the proposed method in ODGEN. As shown in Tab. 12, ODGEN achieves better results on most benchmarks than generating samples with isolated categories. This is an expected result since generating objects of various categories in one image should be better aligned with the test set.

Table 11: Ablations of the number of synthetic samples (Metric: mAP@.50 (↑) of YOLOv5s / YOLOv7, real images # 200). Results are very close between 5k and 10k synthetic samples.

| synth # | Robomaster | Cotton | Aquarium |
|---|---|---|---|
| 0 | 50.3 / 48.8 | 29.4 / 32.0 | 53.8 / 53.0 |
| 1000 | 52.2 / 61.7 | 44.0 / 51.3 | 55.0 / 63.8 |
| 3000 | 61.9 / 61.1 | 62.6 / 56.4 | 56.8 / 65.8 |
| 5000 | **67.4 / 65.4** | **63.9** / 55.0 | 62.6 / **66.5** |
| 10000 | **67.4** / 65.2 | 64.4 / **56.5** | **62.8** / 66.3 |

Table 12: Ablations of category isolation during datasets synthesis (Metric: mAP@.50 (↑) of YOLOv5s / YOLOv7, real + synth images # 200 + 5000). It hinders the performance in most cases.

| Split Categories | True | False |
|---|---|---|
| Road Traffic | 63.4 / 70.3 | **66.8 / 70.4** |
| Aquarium | 57.2 / 63.3 | **62.6 / 66.5** |
| Underwater Object | **41.1** / 42.0 | 40.0 / **44.8** |

Table 13: mAP@.50:.95 (↑) of YOLOv5s / YOLOv7 on RF7. Baseline models are trained with 200 real images only, whereas the other models are trained with 200 real + 5000 synthetic images from various methods. **The corrupted label filtering step is not applied to all methods.** ODGEN still leads to the biggest improvement on all 7 domain-specific datasets.

| | Baseline | ReCo | GLIGEN | ControlNet | GeoDiffusion | ODGEN |
|---|---|---|---|---|---|---|
| real + synth # | 200 + 0 | 200 + 5000 | 200 + 5000 | 200 + 5000 | 200 + 5000 | 200 + 5000 |
| Apex Game | 38.3 / 47.2 | 25.0 / 31.5 | 24.8 / 32.5 | 33.8 / 42.7 | 29.2 / 35.8 | **39.8 / 52.6** |
| Robomaster | 27.2 / 26.5 | 18.2 / 27.9 | 19.1 / 25.0 | 24.4 / 32.9 | 18.2 / 22.6 | **39.0 / 33.3** |
| MRI Image | 37.6 / 27.4 | 42.7 / 38.3 | 32.3 / 25.9 | 44.7 / 37.2 | 42.0 / 38.9 | **46.1 / 41.5** |
| Cotton | 16.7 / 20.5 | 29.3/ 37.5 | 28.0 / 39.0 | 22.6 / 35.1 | 30.2 / 36.0 | **40.5 / 42.1** |
| Road Traffic | 35.3 / 41.0 | 22.8 / 29.3 | 22.2 / 29.5 | 22.1 / 30.5 | 17.2 / 29.4 | **38.2 / 43.2** |
| Aquarium | 30.0 / 29.6 | 23.8 / 34.3 | 24.1 / 32.2 | 18.2 / 25.6 | 21.6 / 30.9 | **32.0 / 38.4** |
| Underwater | 16.7 / 19.4 | 13.7 / 15.8 | 14.9 / 18.5 | 15.5 / 17.8 | 13.8 / 17.2 | **18.9 / 21.6** |

**Corrupted label filtering.** We design this step to filter some labels when objects are not generated successfully, which may be caused by some unreasonable boxes obtained with the pipeline in Fig. 3. For experiments on COCO, this step is not applied to any method since we directly use labels from the COCO validation set and hope to synthesize images consistent with the real-world labels. For RF7 experiments, this step is only applied to our ODGEN in Tab. 2. For fair comparison, we skip this step on RF7 to compare the generation capability of different methods fairly and provide results in Tab. 13. It shows that the corrupted label filtering step only contributes a small part of the improvement. Without this step, our method still outperforms the other methods significantly. In addition, as illustrated in Appendix A, the current dataset synthesis pipeline is designed to compare different methods and can be improved further in future work.

**Foreground region enhancement on COCO.** We set $\gamma$ as 10 in Eq. (2) in our experiments trained on COCO. We add ablations with $\gamma$ value of 0 to ablate foreground region enhancement on COCO and provide quantitative results in Tab. 14. It shows that the foreground region enhancement helps ODGEN achieve better results, especially the layout-image consistency reflected by mAP results.

Table 14: FID (↓) and mAP (↑) of YOLOv5s / YOLOv7 on COCO. FID is computed with 41k synthetic images. For mAP, YOLO models are trained from scratch on 10k synthetic images and validated on 31k real images. Foreground region enhancement contributes to the improvement of both FID and mAP results.

| Method | FID (↓) | mAP@.50 (↑) | mAP@.50:95 (↑) |
|---|---|---|---|
| ODGEN w/ foreground region enhancement | **16.16** | **18.90 / 24.40** | **9.70 / 14.20** |
| ODGEN w/o foreground region enhancement | 16.45 | 16.90 / 23.10 | 8.72 / 13.50 |

**Fine-tuning on both cropped foreground objects and entire images.** For specific domains, taking the Robomaster dataset in RF7 as an example, most images in the dataset contain multiple categories of objects like "armor", "base", "watcher", and "car", which are either strange for Stable Diffusion or different from what Stable Diffusion tends to generate with the same text prompts. For models fine-tuned on entire images only, they cannot obtain direct guidance on which parts in entire images correspond to these objects. As a result, the fine-tuned models cannot synthesize correct images for foreground objects given text prompts like "a base in a screen shot of the Robomaster game". We provide FID results of foreground objects synthesized by models fine-tuned on both entire images and cropped foreground objects (text prompts are composed of object names and the scene name,

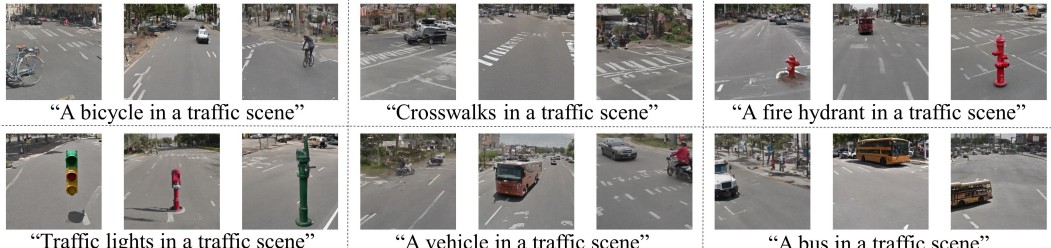

Figure 8: Visualized examples produced by the model fine-tuned on entire images from the Road Traffic dataset only. It fails to produce images of foreground objects correctly.

e.g., "a car and a base in a screen shot of the Robomaster game") and models fine-tuned on entire images only in Tab. 15. It shows that the proposed approach helps fine-tuned models generate images of foreground objects significantly better. Similarly, we also provide FID results of the Road Traffic dataset in Tab. 16, which is relatively more familiar for Stable Diffusion than the Robomaster dataset. Results also show the improvement provided by our approach compared with models fine-tuned on entire images only. Visualized examples produced by the model fine-tuned on entire images only are shown in Fig. 8. Such models struggle to capture the information of foreground objects and fail to produce images needed to build image lists for our ODGEN.

Table 15: FID ($\downarrow$) results of foreground objects in the Robomaster dataset synthesized by models fine-tuned on different data.

| Object categories | watcher | armor | car | base | rune |
|---|---|---|---|---|---|
| Entire images only | 384.94 | 532.56 | 364.46 | 325.79 | 340.11 |
| Entire images and cropped objects | **124.45** | **136.27** | **135.66** | **146.50** | **128.95** |

Table 16: FID ($\downarrow$) results of foreground objects in the Road Traffic dataset synthesized by models fine-tuned on different data.

| Object categories | vehicle | traffic light | motor-cycle | fire hydrant | cross-walk | bus | bicycle |
|---|---|---|---|---|---|---|---|
| Entire images only | 241.64 | 240.55 | 213.90 | 205.22 | 253.40 | 238.63 | 153.76 |
| Entire images and cropped objects | **105.27** | **90.71** | **143.30** | **53.05** | **97.33** | **118.58** | **65.61** |

# D  Implementation Details

## D.1  Global Text Prompt

All the methods included for comparison in our experiments share the same textual descriptions of the foreground objects and entire scenes in the global text prompts for a fair comparison. ReCo [61] and GeoDiffusion [5] further add the location information of foreground objects to their global text prompts following their approaches. For general domains, we concatenate the object class names into a sentence as the global text prompt. For specific domains, we concatenate the object class names and the scene name into a sentence as the global text prompt.

## D.2  ODGEN (ours)

We follow the default training setting used by ControlNet [62], including the learning rates and the optimizer to train our ODGEN models and provide fair comparison with ControlNet.

**Offline foreground objects synthesis.**  To improve the training efficiency, we synthesize 500 foreground object images for each category offline and build a synthetic foreground pool. During the training of object-wise conditioning modules, foreground images are randomly sampled from the pool to build the image list.

**Image encoder.** ControlNet uses a 4-layer convolutional encoder to encode input conditions sized 512 to $64 \times 64$ latents. The input channels are usually set as 1 or 3 for input images. Our ODGEN

follows similar methods to convert the input conditions. The key difference is that we use image lists of synthesized objects as input conditions. The image lists are padded to $N$ and concatenated in the channel dimension for encoding. In this case, we expand the input channels of the encoder to $3N$ and update the channels of the following layers to comparable numbers. We provide the channels of each layer in Tab. 17. The convolutional kernel size is set as 3 for all layers. We design different encoder architectures according to the maximum object numbers that can be found in a single image. For datasets like MRI in which most images contain only one object, we can use a smaller encoder to make the model more lightweight. For a generic model trained on large-scale datasets, we can fix the architectures of encoders with a large $N$ to make it applicable to images containing many objects.

Table 17: Detailed input and output channels of each convolutional layer in the image encoder used in ODGEN. As the maximum object number per image for different datasets varies, we select different $N$ and thus different channel numbers.

| Datasets | $N$ | Layer 1 | | Layer 2 | | Layer 3 | | Layer 4 | |
|---|---|---|---|---|---|---|---|---|---|
| | | Input | Output | Input | Output | Input | Output | Input | Output |
| Apex Game | 6 | 18 | 32 | 32 | 96 | 96 | 128 | 128 | 256 |
| Robomaster | 17 | 51 | 64 | 64 | 96 | 96 | 128 | 128 | 256 |
| MRI Image | 2 | 6 | 16 | 16 | 32 | 32 | 96 | 96 | 256 |
| Cotton | 27 | 81 | 96 | 96 | 128 | 128 | 192 | 192 | 256 |
| Road Traffic | 19 | 57 | 64 | 64 | 96 | 96 | 128 | 128 | 256 |
| Aquarium | 56 | 168 | 168 | 168 | 192 | 192 | 224 | 224 | 256 |
| Underwater | 79 | 237 | 237 | 237 | 256 | 256 | 256 | 256 | 256 |
| COCO | 93 | 297 | 297 | 297 | 256 | 256 | 256 | 256 | 256 |

**Text embedding encoder.** Similar to the image list, the text list is also padded to the length $N$ with empty text. Each item in the text list in encoded by the CLIP text encoder to an embedding. With the CLIP text encoder used in Stable Diffusion 1.x and 2.x models, we get $N$ text embeddings sized $(\text{batch size}, 77, 1024)$. Then we stack the embeddings to size $(\text{batch size}, N, 77, 1024)$ and use a 4-layer convolutional layer to compress the information into one embedding. The input channel is $N$ and the output channels of following layers are $\lfloor N/2 \rfloor, \lfloor N/4 \rfloor, \lfloor N/8 \rfloor, 1$. We set the convolutional kernel size as 3, the stride as 1, and use zero padding to maintain the sizes of the last two dimensions.

**Foreground/Background discriminator.** We train discriminators by fine-tuning ImageNet pre-trained ResNet50 on foreground and background patches randomly cropped from training sets. They are split for training, validating, and testing by 70%, 10%, and 20%. The model is a binary classification model that only predicts whether a patch contains any object. The accuracy on test datasets is over 99% on RF7. Therefore, we can confidently use it to filter the pseudo labels.

### D.3   Other Methods

**ReCo [61] & MIGC [67] & GeoDiffusion [5]** are layout-to-image generation methods based on Stable Diffusion. For the comparison on RF7, we use the official codes of ReCo and GeoDiffusion to fine-tune the diffusion models on domain-specific datasets and generate synthetic datasets for experiments. For the comparison on COCO, we directly use their released pre-trained weights trained on COCO. MIGC only releases the pre-trained weight but not the code, so we only compare it with our method on the COCO benchmark. ReCo and GeoDiffusion fine-tune both the text encoder and the UNet of Stable Diffusion. MIGC only fine-tunes the UNet.

**InstanceDiffusion [55]** depends on multiple formats of inputs, including segmentation masks, boxes, and scribbles, to train its UniFusion module. However, the RF7 datasets of specific domains do not provide segmentation masks and scribbles. Therefore, it is only employed for comparison on COCO with its open-source checkpoint.

**GLIGEN [33]** is an open-set grounded text-to-image generation methods supporting multiple kinds of conditions. We use GLIGEN for layout-to-image generation with bounding boxes as conditions. GLIGEN is designed to fit target distributions by only fine-tuning the inserted gated self-attention layers in pre-trained diffusion models. Therefore, we fix the other layers in the diffusion model during fine-tuning on RF7 or COCO.

**ControlNet [62]** is initially designed to control diffusion models by fixing pre-trained models, reusing the diffusion model encoder and fine-tuning it to provide control. In our experiments, to make fair comparisons, we also fine-tune the diffusion model part to fit it to specific domains that can be different from the pre-training data of Stable Diffusion. The diffusion model remains trainable for experiments on COCO as well. The native ControlNet is originally designed as an image-to-image generation method. As illustrated in Sec. 4, we provide a simple way to convert boxes to masks. Given a dataset containing $K$ categories, an image sized $H \times W$, and $B$ objects with their bounding boxes $bbox$ and classes $cls$, we convert them to a mask $M$ in the following process:

---

**Algorithm 1** Convert bounding boxes to a mask for ControlNet

---

$M = \text{zeros}(H, W, K)$
**for** $i = 1, \ldots, B$ **do**
    $coord_{left}, coord_{top}, coord_{right}, coord_{bottom} = bbox[i]$
    $k = cls[i]$
    $M[coord_{top} : coord_{bottom}, coord_{left} : coord_{right}, k]+ = 1$
**end for**

---

Table 18: The number of images in the 7 subsets of Roboflow-100 used to compose the RF7 datasets.

| Datasets | Train | Validation | Test |
|---|---|---|---|
| Apex Game | 2583 | 415 | 691 |
| Robomaster | 1945 | 278 | 556 |
| MRI Image | 253 | 39 | 79 |
| Cotton | 367 | 20 | 19 |
| Road Traffic | 494 | 133 | 187 |
| Aquarium | 448 | 63 | 127 |
| Underwater | 5320 | 760 | 1520 |

## E  Datasets Details

**RF7 datasets.** We employ the RF7 datasets from Roboflow-100 [6] to evaluate the performance of our ODGEN on specific domains. Here, we add more details of the employed datasets, including Apex Game, Robomaster, MRI Image, Cotton, Road Traffic, Aquarium, and Underwater. We provide the number of images for each dataset, including the train, validation, and test sets in Tab. 18. We use the first 200 samples in the image list annotation provided by Roboflow-100 for each dataset to build the training set for our experiments. The full validation set is used as the standard to choose the best checkpoint from YOLO models trained for 100 epochs. The full test set is used to evaluate the chosen checkpoint and provide mAP results shown in this paper. Visualized examples with annotations are shown in Fig. 9. The object categories of each dataset are provided in Tab. 19. Some images in the Apex Game and Robomaster datasets contain complex and tiny text that are hard to generate by current Stable Diffusion.

## F  Supplemental Related Works

Copy-paste [14, 65] method requires segmentation masks to get the cropped foreground objects, which are not provided by the RF7 datasets used in this paper. Besides, our approach also serves as a controllable image generation method that only needs bounding box labels to produce realistic images, which is hard to achieve by copy-paste. InstaGEN [11] is mainly designed for the open-vocabulary object detection task. It generates images randomly and annotates them with a grounding head trained with another pre-trained detector. Earlier works like LayoutDiffusion [66] are not implemented with latent diffusion models and thus are not included for comparison. Works [23, 39, 56] are designed for semantic segmentation, while our ODGEN is designed for object detection.

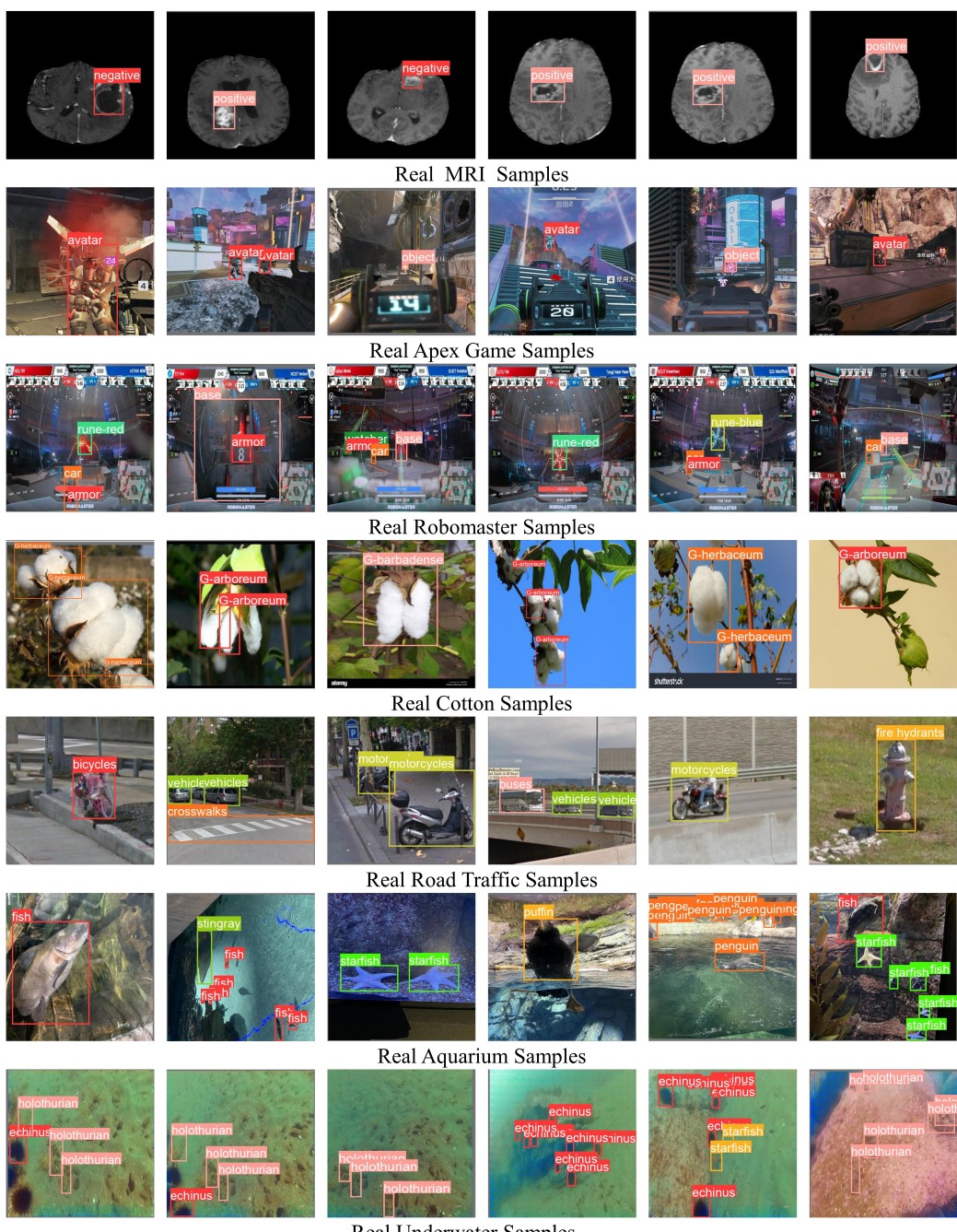

Figure 9: Visualized examples with object detection annotations of RF7 datasets.

# G Supplemental Experiments

**mAP@.50 on RF7.** Tab. 20 provides quantitative results for trainability in terms of mAP@.50 as supplements to Tab. 2. Our approach achieves improvement over training on real data only and outperforms other methods.

**Visual relationship between image lists and synthesized images.** The image list is designed to provide information on category and localization for controlling foreground object layouts. The synthesized images share the same category with objects in generated images and are pasted to

| Datasets | Class Names |
|---|---|
| Apex Game | Avatar, Object |
| Robomaster | Armor, Base, Car, Rune, Rune-blue, Rune-gray, Rune-grey, Rune-red, Watcher |
| MRI Image | Negative, Positive |
| Cotton | G-arboreum, G-barbadense, G-herbaceum, G-hirsitum |
| Road Traffic | Bicycles, Buses, Crosswalks, Fire hydrants, Motorcycles, Traffic lights, Vehicles |
| Aquarium | Fish, Jellyfish, Penguin, Puffin, Shark, Starfish, Stingray |
| Underwater | Echinus, Holothurian, Scallop, Starfish, Waterweeds |

Table 19: Class names of the RF7 datasets.

Table 20: mAP@.50 (↑) of YOLOv5s / YOLOv7 on RF7. Baseline models are trained with 200 real images only, whereas the other models are trained with 200 real + 5000 synthetic images from various methods. ODGEN leads to the biggest improvement on all 7 domain-specific datasets.

| | Baseline | ReCo | GLIGEN | ControlNet | GeoDiffusion | ODGEN |
|---|---|---|---|---|---|---|
| real + synth # | 200 + 0 | 200 + 5000 | 200 + 5000 | 200 + 5000 | 200 + 5000 | 200 + 5000 |
| Apex Game | 64.4 / 78.5 | 53.3 / 60.0 | 53.3 / 57.0 | 59.3 / 70.3 | 55.8 / 62.1 | **69.1 / 82.6** |
| Robomaster | 50.3 / 48.8 | 41.6 / 54.4 | 38.7 / 50.2 | 44.4 / 57.0 | 41.5 / 45.7 | **67.4 / 65.4** |
| MRI Image | 55.1 / 44.7 | 59.7 / 59.6 | 52.1 / 42.0 | 64.7 / 55.4 | 61.6 / 56.1 | **68.1 / 61.3** |
| Cotton | 29.4 / 32.0 | 42.7 / 53.4 | 41.8 / **55.0** | 38.3 / 47.1 | 49.6 / 49.4 | **63.9 / 55.0** |
| Traffic | 61.1 / 68.8 | 44.6 / 52.4 | 45.0 / 54.5 | 43.5 / 56.2 | 38.2 / 54.6 | **66.8 / 70.4** |
| Aquarium | 53.8 / 53.0 | 45.8 / 59.0 | 49.7 / 51.2 | 35.5 / 45.4 | 45.1 / 54.8 | **62.6 / 66.5** |
| Underwater | 35.7 / 39.2 | 30.9 / 34.5 | 33.6 / 38.7 | 34.3 / 37.7 | 31.8 / 37.6 | **40.0 / 44.8** |

the same position as those in generated images to build image lists. The objects in image lists and synthesized images may all be apples or cakes but have different shapes and colors, as shown by the visualized samples provided in Fig. 10.

**Detector performance improvement with synthetic data.** We provide several visualized samples in Fig. 11 and show that the synthetic data helps detectors detect some occluded objects.

**Generalization to novel categories.** We use ODGEN trained on COCO to synthesize samples containing subjects not included in COCO, like moon, ambulance, tiger, et al. We show visualized samples in Fig. 12. It shows that ODGEN trained on COCO can control the layout of novel categories. However, we find that the generation quality is not very stable, which may be influenced by the fine-tuning process on the COCO dataset. In future work, we hope to get a powerful and generic model that is robust to the most common categories in daily life.

**Computational cost.** Our ODGEN needs to synthesize images of foreground objects to build image lists, leading to higher computational costs for training and inference. Therefore, we propose to generate an offline image library of foreground objects to accelerate the process of building image lists. With the offline library, we can randomly pick images from it to build image lists instead of synthesizing new images of foreground objects every time.

Taking the model trained on the COCO dataset as an example, our ODGEN shares a very close scale of parameters with ControlNet (Trainable parameters: ODGEN 1231M v.s. ControlNet 1229M, parameters in the UNet of Stable Diffusion are included). We provide the training time for 1 epoch on COCO with 8 V100 GPUs of different methods in Tab. 21.

Table 21: Training time for 1 epoch on COCO with 8 V100 GPUs.

| Method | ReCo | GLIGEN | ControlNet | GeoDiffusion | ODGEN |
|---|---|---|---|---|---|
| Training Time | 3 hours | 7 hours | 4.5 hours | 3.2 hours | 5.6 hours |

In the inference stage (Fig. 2(c)), our ODGEN pads both the image and text lists to a fixed length. Therefore, the computational cost for inference with an offline library doesn't increase significantly with more foreground objects. Other methods like InstanceDiffusion [55] and MIGC [67] need more time for training and inferencing with more objects. Taking models trained on COCO as examples, to generate an image with 6 bounding boxes on a V100 GPU, ODGEN takes 10 seconds, ControlNet takes 8 seconds, and InstanceDiffusion takes 30 seconds.

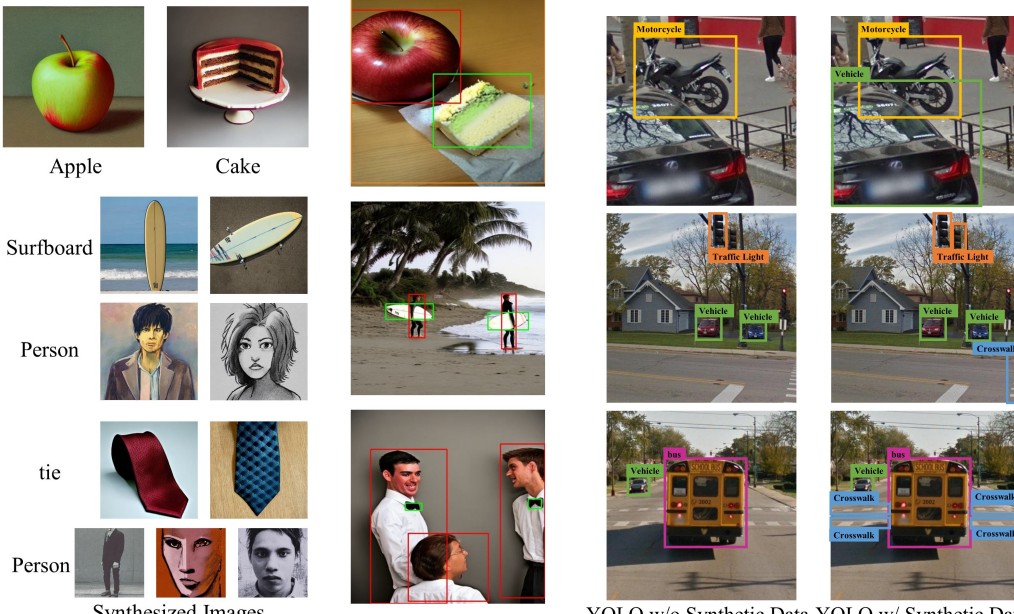

Figure 10: Left: images used to build image lists. Right: corresponding synthesized samples. The synthesized images of foreground objects share only the same category with objects in synthesized samples but not colors or shapes.

Figure 11: Detection results of YOLO detectors trained w/ or w/o ODGEN synthetic data. The synthesized samples of ODGEN help detectors perform better on partially occluded objects.

During the inference process, we compare using the same offline image library as training against using a totally different image library. We get very close results, as shown in Tab. 22. Both outperform other methods significantly, as shown in Tab. 4. It indicates that ODGEN can extract category information from synthesized samples of foreground objects in image lists instead of depending on certain images of foreground objects. ODGEN is not constrained by the image library of foreground objects used in training and can be generalized to other newly generated offline image libraries consisting of novel synthesized samples of foreground objects, which ensures that offline image libraries increase ODGEN's computational efficiency without reducing its practicality.

Table 22: FID (↓) and mAP (↑) of YOLOv5s / YOLOv7 for ODGEN trained on COCO. FID is computed with 41k synthetic images. For mAP, YOLO models are trained from scratch on 10k synthetic images and validated on 31k real images. Different offline image libraries are applied for inference.

| Offline Image Library | FID (↓) | mAP@.50 (↑) | mAP@.50:95 (↑) |
|---|---|---|---|
| Same as training | 16.01 | 18.90 / 9.70 | 24.40 / 14.20 |
| Different from training | 16.16 | 18.60 / 9.52 | 24.20 / 14.10 |

**Quantitative evaluation for concept bleeding.** This work mainly focuses on the concept of the bleeding problem of different categories of objects. The mAP results of YOLO models trained on synthetic images only in Tab. 4 can be a proxy task to prove that the concept bleeding is alleviated since our ODGEN achieves state-of-the-art performance in the layout-image consistency of complex scene generation conditioned on bounding boxes. This section adds quantitative evaluation with BLIP-VQA [21], which employs the BLIP [32] model to identify whether the contents in synthesized images are consistent with text prompts. The results are provided in Tab. 23. Our ODGEN outperforms other methods and gets results close to ground truth (real images from the COCO validation set sharing the same labels with synthetic images).

**ODGEN trained on COCO with Stable Diffusion v1.5.** Our ODGEN is implemented with Stable Diffusion v2.1 in Sec. 4. We add experiments of implementing ODGEN with Stable Diffusion v1.5

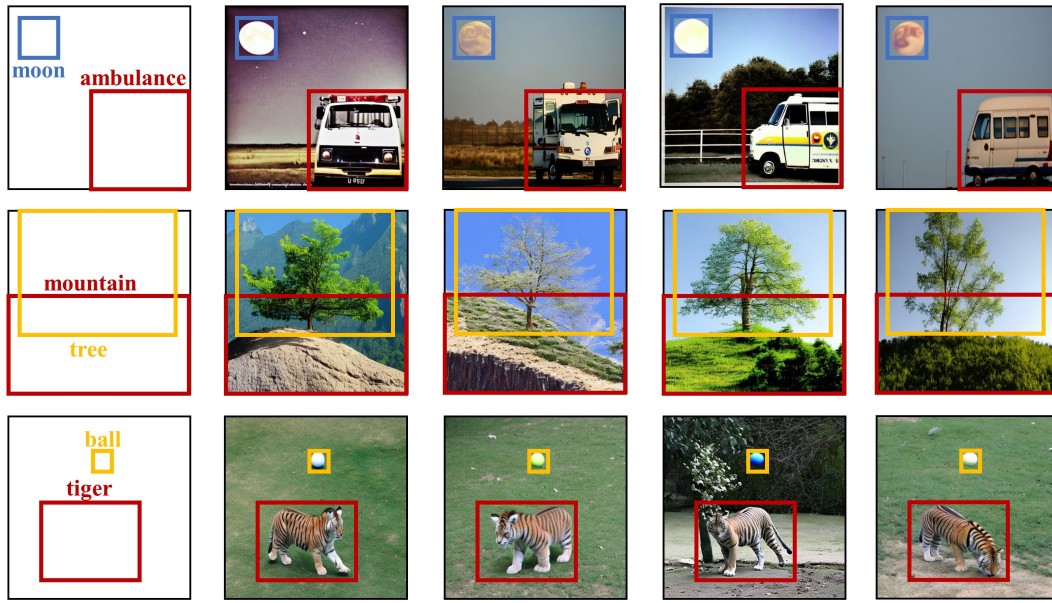

Figure 12: Visualized samples containing novel categories of foreground objects generated by ODGEN trained on the COCO dataset. Stable Diffusion is used to generate images of the foreground objects to build image lists for inference. It shows that our ODGEN can control the layout of novel categories that were never seen in its training process.

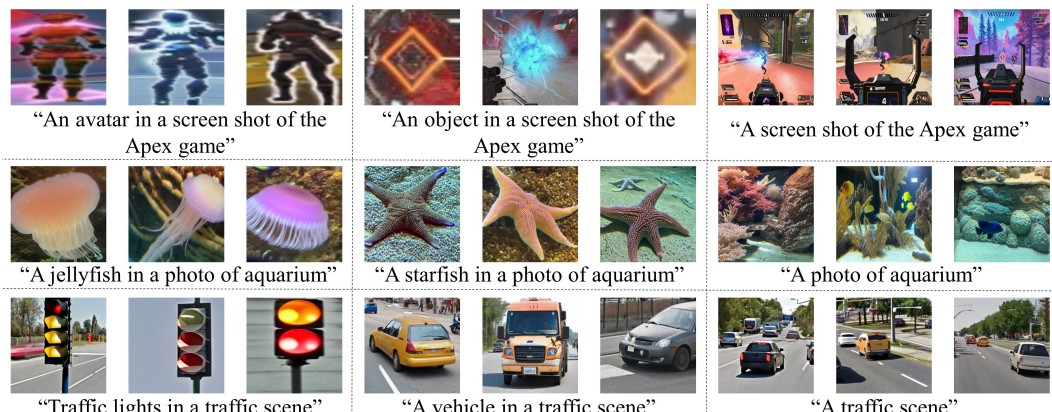

Figure 13: Visualized samples generated by the models fine-tuned on both cropped foreground objects and entire images from RF7 datasets (200 images for each dataset). Text prompts are provided under figures. Columns 1-6 are generated foreground objects and columns 7-9 are generated entire images. Models fine-tuned on entire images only cannot synthesize foreground objects required by image lists since they cannot bind the prompts to corresponding objects in entire images that may contain multiple categories of objects.

trained on COCO for comparison in this part. Results are provided in Tab. 24. Our ODGEN achieves similar results with different versions of Stable Diffusion models.

**Visualization.** We provide extensive visualized examples of ODGEN to demonstrate its effectiveness. In Fig. 13, we provide visualized samples produced by models fine-tuned on both cropped foreground objects and entire images. It shows that the fine-tuned models are capable of generating diverse foreground objects and complete scenes. In Fig. 14, we show generated samples of the specific domains in RF7. It can be seen that ODGEN is robust to complex scenes composed of multiple categories, dense and overlapping objects. Besides, ODGEN shows strong generalization capability to various domains.

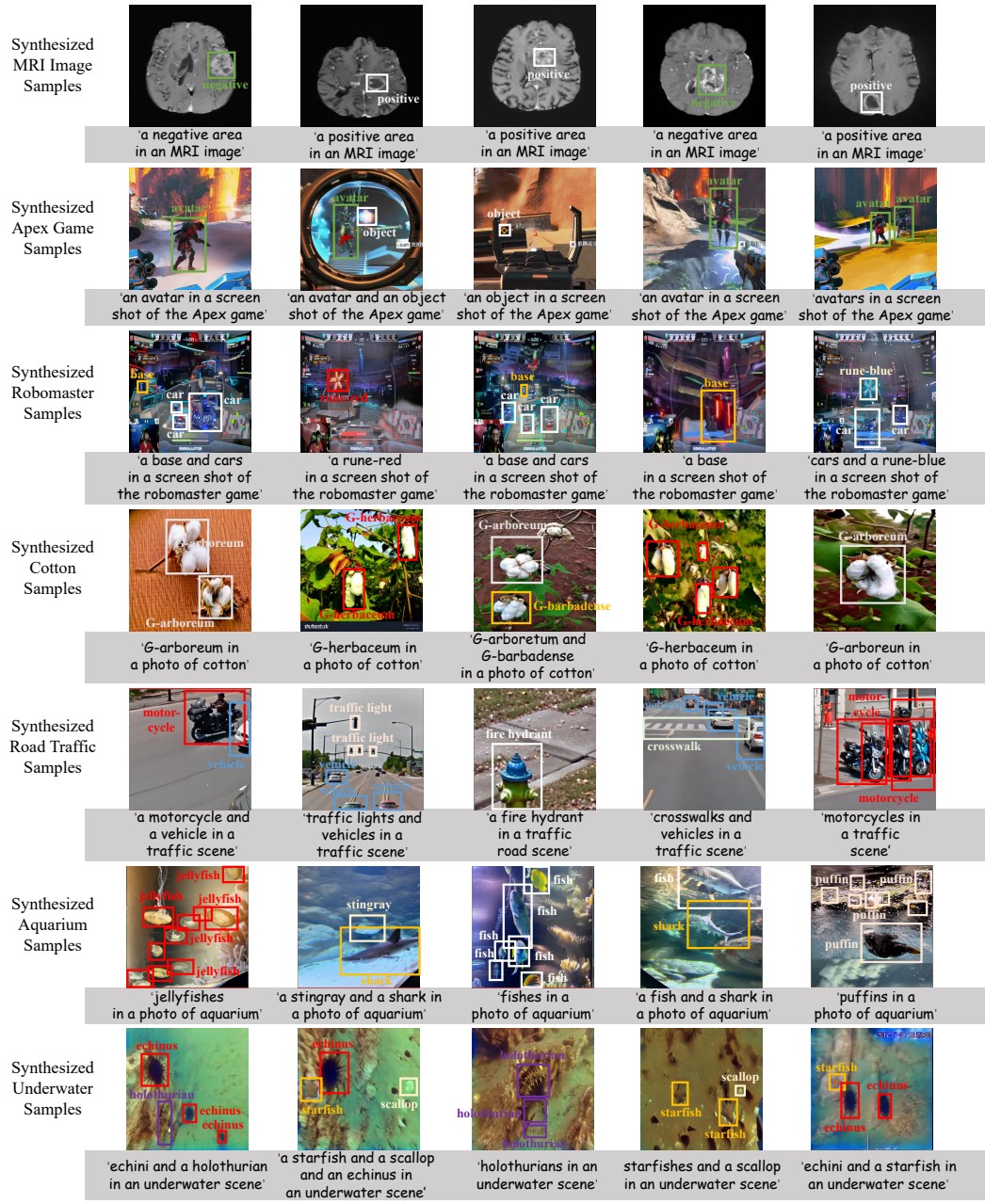

Figure 14: Additional visualized samples produced by our approach in various specific domains. Annotations of bounding boxes are marked on samples and global text prompts are provided below corresponding samples. Our approach can be generalized to various domains and is compatible with complex scenes with multiple objects, multiple categories, and bounding box occlusions.

In Fig. 15, Fig. 16, and Fig. 17, we provide supplemental visualized comparisons between ODGEN and the other methods on the COCO-2014 dataset. ODGEN achieves the best image-annotation consistency and maintains high generation fidelity. More generated samples of ODGEN are shown in Fig. 18. In addition, we also show different examples produced by ODGEN on the same conditions in Fig. 19 to show its capability of generating diverse results.

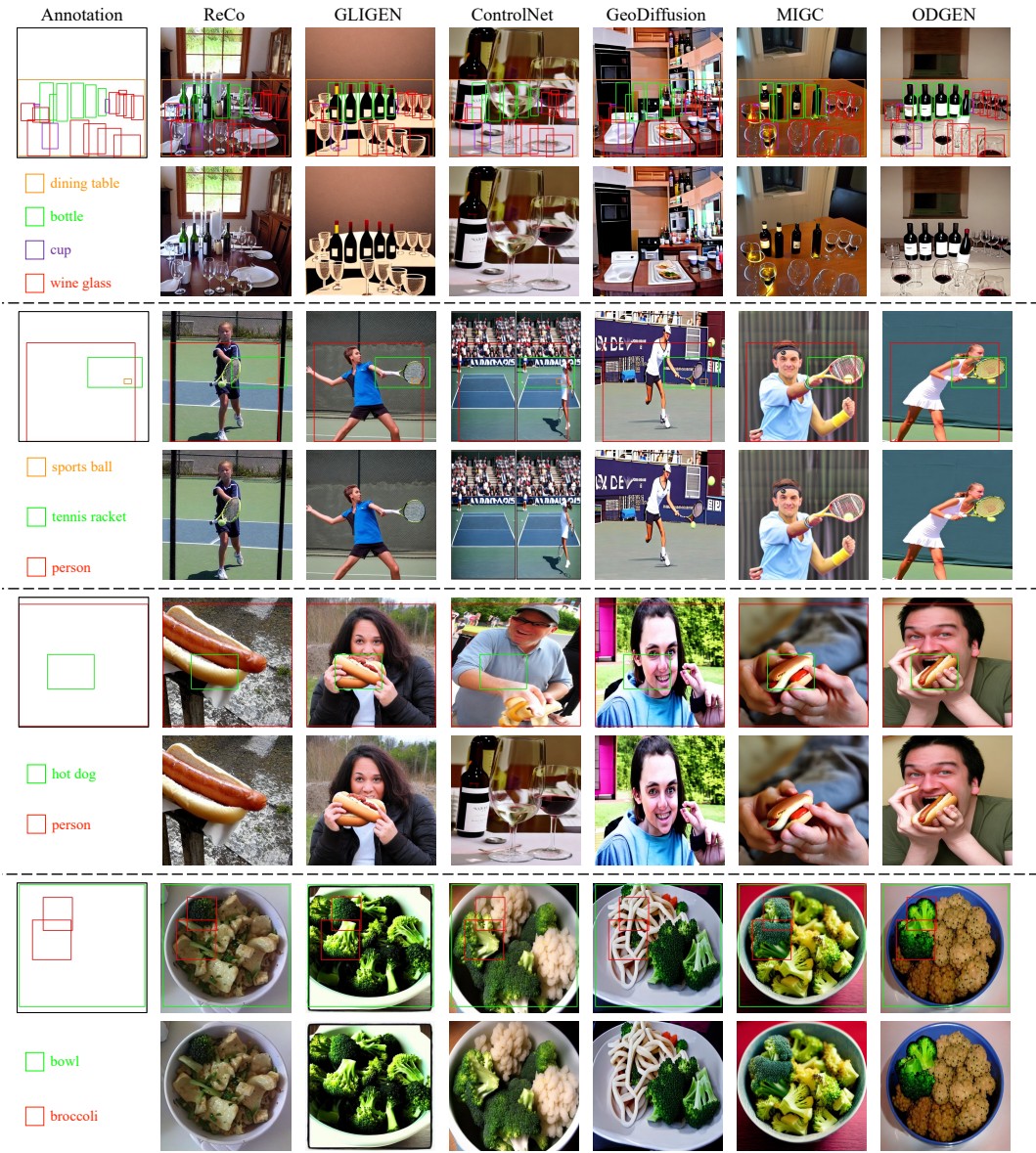

Figure 15: Visualized comparison results on the COCO-2014 dataset. The annotations used as conditions to synthesize images are shown in the first column.

Table 23: BLIP-VQA (↑) results averaged over 41k images synthesized with different methods using the labels from the COCO validation set.

| Method | ReCo | GLIGEN | ControlNet | GeoDiffusion |
|---|---|---|---|---|
| BLIP-VQA (↑) | 0.2027 | 0.2281 | 0.2461 | 0.2114 |
| Method | MIGC | InstanceDiffusion | ODGEN | Ground Truth (reference) |
| BLIP-VQA (↑) | 0.2314 | 0.2293 | **0.2716** | 0.2745 |

Table 24: FID (↓) and mAP (↑) of YOLOv5s / YOLOv7 on COCO. FID is computed with 41k synthetic images. For mAP, YOLO models are trained from scratch on 10k synthetic images and validated on 31k real images.

| Method | FID (↓) | mAP@.50 (↑) | mAP@.50:95 (↑) |
|---|---|---|---|
| ODGEN based on Stable Diffusion v2.1 | 16.16 | **18.90** / 24.40 | **9.70** / 14.20 |
| ODGEN based on Stable Diffusion v1.5 | **15.93** | 18.20 / **24.50** | 9.39 / **14.30** |

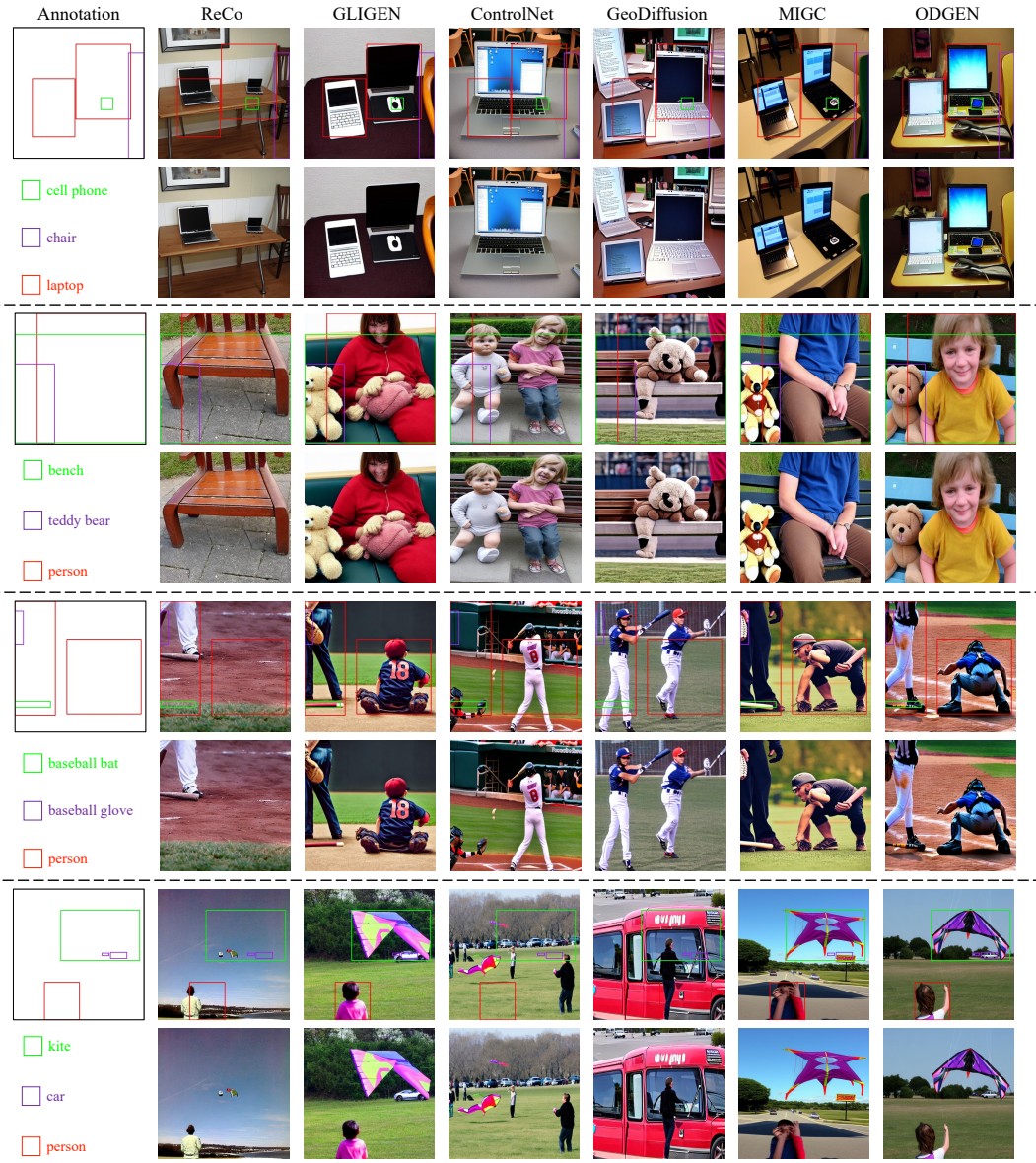

Figure 16: Visualized comparison results on the COCO-2014 dataset. The annotations used as conditions to synthesize images are shown in the first column.

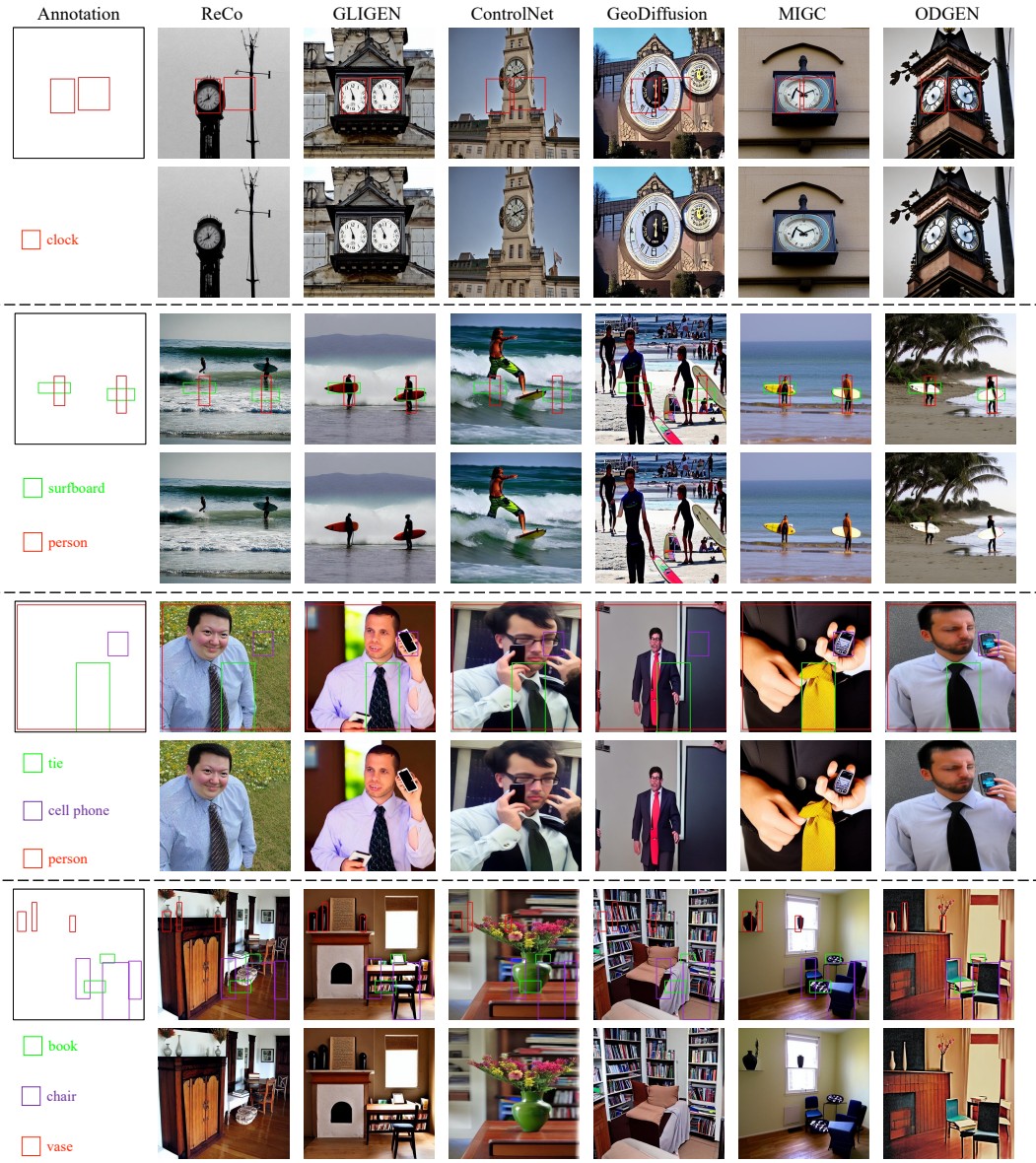

Figure 17: Visualized comparison results on the COCO-2014 dataset. The annotations used as conditions to synthesize images are shown in the first column.

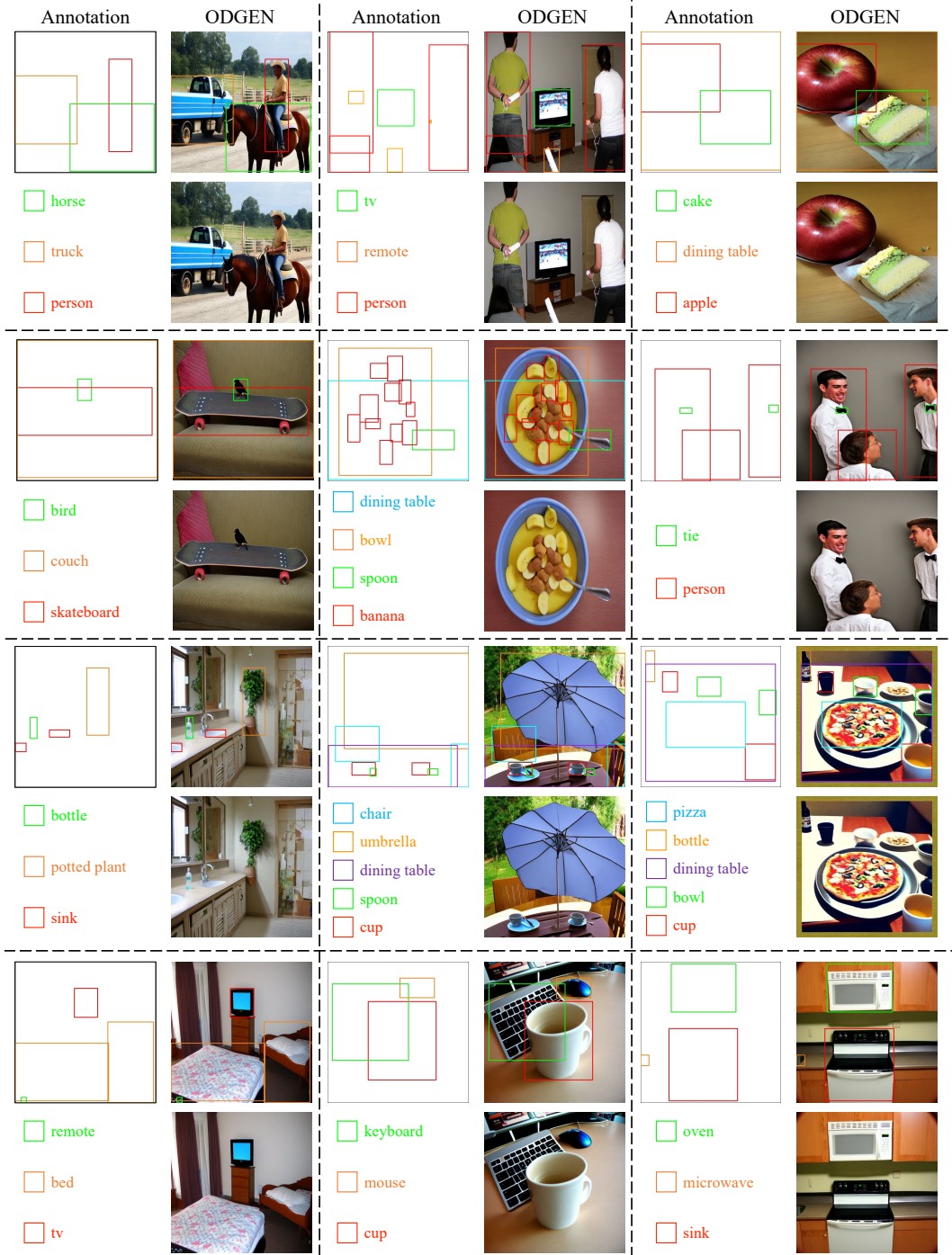

Figure 18: Visualized results of ODGEN on the COCO-2014 dataset. In every grid, the annotation used as conditions to synthesize images are shown in the top-left corner.

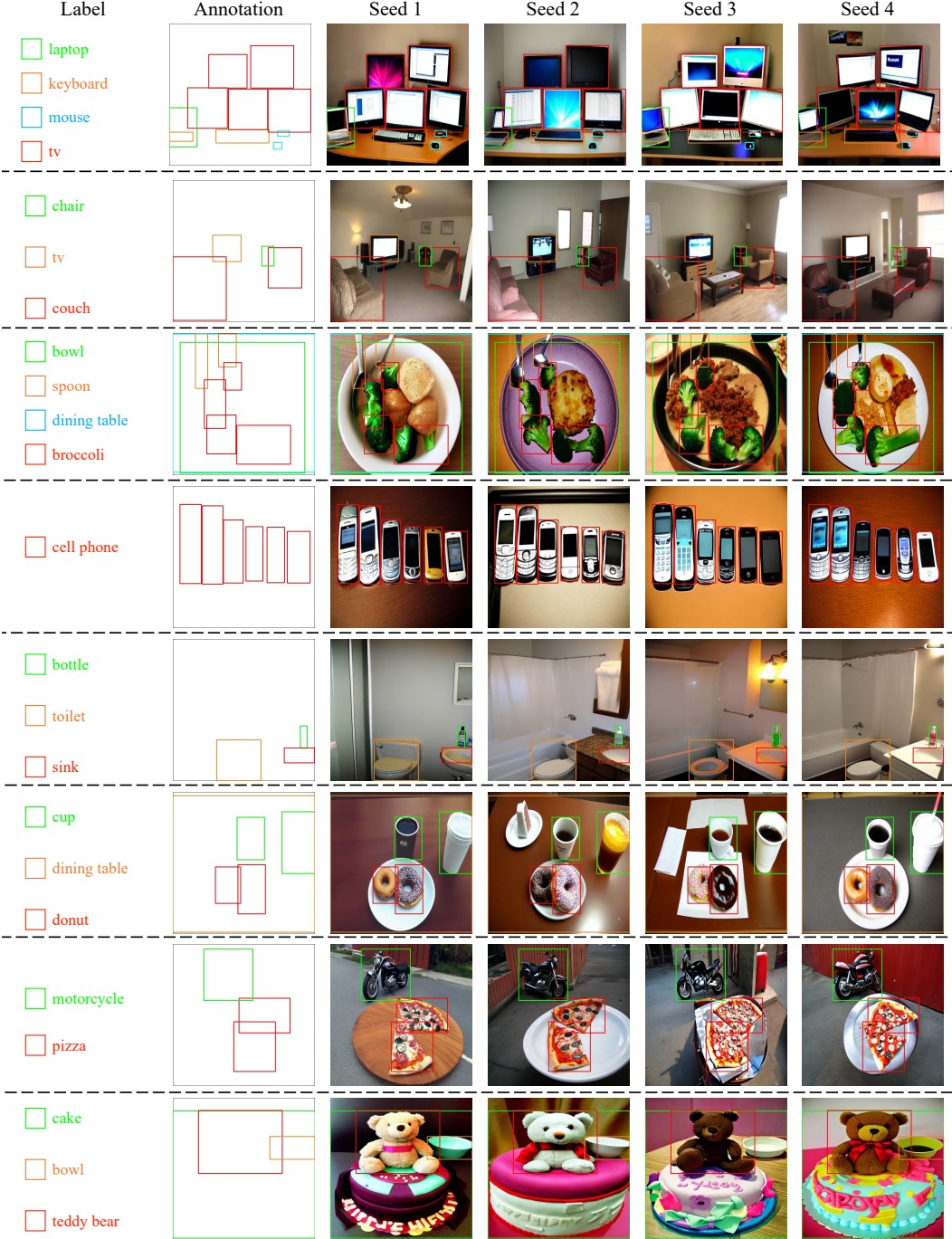

Figure 19: Visualized results of our ODGEN on the COCO-2014 dataset generated from the same conditions. The annotation is placed on the first column.

