# OpenReview forum: "ODGEN: Domain-specific Object Detection Data Generation with Diffusion Models"
_NeurIPS.cc/2024/Conference — NeurIPS 2024 poster_

### Official Review · Reviewer_FcFR · 2024-07-09

**Soundness:** 3
**Presentation:** 3
**Contribution:** 3
**Rating:** 6
**Confidence:** 5

**Summary:**

This paper tackles the task of controllable image generation from bounding boxes and text prompts for training object detectors. To fine-tune diffusion models to specific domains effectively and deal with the challenge of concept bleeding, this paper proposes a method called ODGEN. In the proposed method, the diffusion model is first fine-tuned on both entire images and cropped foreground regions. Then, text lists and image lists are utilized as input for ControlNet to avoid the concept bleeding and generate high-quality images. The experimental results show that the proposed method outperforms the state-of-the-art methods for controllable image generation on seven types of datasets in terms of FID score. In addition, the object detectors trained on the generated images from the proposed method achieve better performance than other methods.

**Strengths:**

i) The proposed method is quite simple yet effective. Its relative ease of implementation is helpful for computer vision practitioners and future researchers. Also, the task tackled in this paper is of practical benefit.

ii) This paper is well-written and easy to follow. The motivation for tackling the task is clearly described in Sec. 1, and related works are well-summarized in Sec. 2. The proposed method is clearly explained at both the idea level and procedure level in Sec. 3.

iii) The experiments demonstrate the high performance of the proposed method. The proposed method achieved better FID scores than other controllable image generation methods on seven types of object detection datasets. Additionally, training with synthetic images from the proposed method improved the performance of YOLO-based object detectors significantly.

**Weaknesses:**

My concerns are mainly about the lack of detailed analysis:

I) This paper claims that fine-tuning with both cropped foreground regions and entire images is one of its contributions. However, an ablation study to confirm the effectiveness of this approach is missing.

ii) From Fig. 4, it looks like the corrupted label filtering is performed only for the proposed method (ODGEN). Because the corrupted label filtering is a simple technique and can be applied to other methods as well, the comparisons between ODGEN and other methods not only without filtering but also with filtering should be provided for fair comparisons.

iii) This paper provides only the results with 200 real images. I'm curious to know how the performance of the proposed method and other methods change when more number of real images are available for training.

iv) In Sec. 1, the concept bleeding is raised as one of the challenges in this task. Although Fig. 6 provides an example, I would like to see a quantitative evaluation to check whether it was effectively addressed by the proposed method.

**Questions:**

See iii) in Weakness.
How is the performance of the proposed method and other methods when more real images are available?

**Limitations:**

The limitations are adequately discussed in Sec. A.

---

> ### Author Rebuttal · Authors · 2024-08-03
>
> Thanks for your valuable comments. We provide detailed responses below to resolve your concerns.
>
> $\textbf{1. Ablations for fine-tuning on both cropped foreground objects and entire images:}$
>
> The fine-tuning process is the basis of the training of the following object-wise conditioning module. The fine-tuning on both cropped foreground objects and entire images makes the fine-tuned models capable of synthesizing the images of foreground objects, which are needed by building image lists for the next step. We provide some visualized samples in Fig. 1 of the PDF file attached to the global response. It shows that the fine-tuned model is capable of generating diverse foreground objects and complete scenes. The training of the object-wise conditioning module is designed to realize layout control based on the fine-tuned model and uses the image lists containing synthesized images of foreground objects as conditions. Therefore, it's hard to conduct an ablation study that fine-tunes the diffusion model without cropped foregrounds. Because the models only fine-tuned on entire images have difficulties in generating foreground objects required by image lists since they cannot bind the prompts to corresponding objects in entire images that may contain multiple categories of objects.
>
> $\textbf{2. Post-processing step with the corrupted label filtering:}$
>
> See the analysis and experiments provided in part 1 of the global response. We provide results of all methods without post-processing to compare the generation capability directly and fairly.
>
> $\textbf{3. Experiments with a larger number of real images:}$
>
> See the experiments provided in the parts 2 and 3 of the global response. We provide experiments with different methods trained on the COCO training set (80k images) and add experiments of our ODGEN trained on 1000 images from the Apex Game and Underwater Object datasets.
>
> $\textbf{4. Quantitative evaluation of concept bleeding:}$
>
> This work mainly focuses on the concept bleeding problem of different categories. The mAP of YOLO models trained on synthetic images only in Tab. 3 of our paper can be a proxy task to prove that concept bleeding is alleviated since our approach achieves state-of-the-art performance in the layout-image consistency of complex scene generation conditioned on bounding boxes. In addition, we add the quantitative evaluation with BLIP-VQA ($\uparrow$) [a] which employs the BLIP model to identify whether the contents in synthesized samples are consistent with the text prompts. The results are shown in the table below. Our ODGEN outperforms other methods and gets results close to ground truth (real images from the COCO dataset sharing the same labels with synthetic images). The results are averaged over 41k synthetic images following the labels of the COCO validation set.
>
> Method | ReCo | GLIGEN | ControlNet | GeoDiffusion | MIGC | InstanceDiffusion | ODGEN (ours) | Ground Truth (provided as reference)
> ------|------|-------|-------|-------|------|-------|------|------
> BLIP-VQA ($\uparrow$) | 0.2027 | 0.2281 | 0.2461 | 0.2114 | 0.2314 | 0.2293 | $\textbf{0.2716}$ | 0.2745
>
> [a] K. Huang, K. Sun, E. Xie, Z. Li, and X. Liu, T2i-compbench: A comprehensive benchmark for open-world compositional text-to-image generation, Advances in Neural Information Processing Systems, vol. 36, 2024.

---

> ### Author Response · Authors · 2024-08-14
>
> Dear Reviewer FcFR,
>
> Thanks for your positive feedback. We are glad that most of your concerns have been well addressed. We add a detailed explanation for your remaining concern.
>
> 1. $\textbf{Specific Domains}$:
>
> Taking the Robomaster dataset in RF7 as an example, most images in the dataset contain multiple categories of objects like "armor", "base", "watcher" and "car",  which are either strange for Stable Diffusion or different from what Stable Diffusion tends to generate with the same text prompts. For the model fine-tuned on entire images only, it cannot obtain guidance on which parts in entire images correspond to these objects. As a result, the fine-tuned models cannot synthesize correct images for foreground objects given text prompts like "a base in a screen shot of the robomaster game". We agree with you that "adding experimental results with the models fine-tuned on entire images only" can make our statement more convincing. We provide FID results of foreground objects synthesized by models fine-tuned on both entire images and cropped foreground objects (text prompts are composed of the names of objects in the image and the scene name, e.g., "a car and a base in a screen shot of the robomaster game"), and models fine-tuned on entire images only, in the table below for comparison:
>
> Table: FID ($\downarrow$) results of foreground objects in the Robomaster dataset synthesized by models fine-tuned on different data.
> Object categories | watcher | armor | car | base | rune
> -------|--------|-------|-------|--------|-------
> Models fine-tuned on entire images only | 384.94 | 532.56 | 364.46 | 325.79 | 340.11
> Models fine-tuned on both cropped objects and entire images | $\textbf{124.45}$ | $\textbf{136.27}$ | $\textbf{135.66}$ | $\textbf{146.50}$ | $\textbf{128.95}$
>
> It shows that our approach helps fine-tuned models generate images of foreground objects better. Similarly, we also provide results on the Road Traffic dataset, which is relatively more familiar for Stable Diffusion than the Robomaster dataset.
>
> Table: FID ($\downarrow$) results of foreground objects in the Road Traffic dataset synthesized by models fine-tuned on different data.
> Object categories | vehicle | traffic light | motorcycle | fire hydrant | crosswalk | bus | bicycle
> -------|--------|-------|-------|--------|-------|-------|-------
> Models fine-tuned on entire images only | 241.64 | 240.55 | 213.90 | 205.22 | 253.40 | 238.63 |153.76
> Models fine-tuned on both cropped objects and entire images | $\textbf{105.27}$ | $\textbf{90.71}$ | $\textbf{143.30}$ | $\textbf{53.05}$ | $\textbf{97.33}$ | $\textbf{118.58}$ | $\textbf{65.61}$
>
> Results also show improvement of our approach compared with models fine-tuned on entire images only. Since links are not allowed in the responses, we cannot provide supplemental visualized results here but will add them to the revised manuscript.
>
> 2. $\textbf{General Domains}$:
>
> The fine-tuning step is designed for specific datasets. As for datasets of general domains like COCO, Stable Diffusion can generate the same categories of objects as the objects in the COCO dataset without fine-tuning. Therefore, we skip the fine-tuning step and directly use Stable Diffusion to synthesize images of foreground objects and use them to build image lists. The results in our paper show superior improvement achieved by our ODGEN compared with prior works.
> ﻿
> Thanks for your valuable suggestions and efforts in reviewing this paper again!
> ﻿
> Authors

---

> > ### Comment · Reviewer_FcFR · 2024-08-14
> >
> > Thanks again for the response. Because my remaining concern has been addressed, I'm happy to raise my rating.

---

> ### Author Response · Authors · 2024-08-14
>
> Dear Reviewer FcFR,
>
> Thank you for your positive feedback and efforts in reviewing this paper again!
>
> Authors

---

### Official Review · Reviewer_2vbs · 2024-07-12

**Soundness:** 3
**Presentation:** 4
**Contribution:** 3
**Rating:** 6
**Confidence:** 4

**Summary:**

This work is about generating synthetic (annotated) data for object detection with diffusion models that are tuned for a specific image domain. Generating synthetic annotated data can be useful for model training in situations where data is scarce. The authors propose a specific pipeline and new modules to generate such data. First, an off-the-shelf diffusion model is fine-tuned on a specific domain, for instance a specifc object detection dataset. This is done both with the full images and crops of foreground objects (via the given bounding boxes). The statistics of object categories, number of instances, bounding boxes are estimated. Then, a new layout is sampled from the distribution, the class-specific objects are individually synethesized with the tuned diffusion model. These are individually placed at their sampled bounding box location to build a so-called image list. Then, a ControlNet model is used that encodes this image list along with a text list of the object categories, which outputs the final synthesized image. The newly designed text and image encoders for the controlnet are trained. Finally, a verification step is done at the object level (classifying if the generated image contains the desired object inside a certain bounding box). The synthesized data is evaluated with the FID metric, and also used to train object detectors, which are then evaluated with mAP.

**Strengths:**

- The problem the paper tries to address is important and has several use cases, maybe even beyond what the paper lists.
- I especially like the adaptation to specific domains, which is certainly relevant for many real-world use cases.
- I enjoyed reading the paper, it's well written, has a good flow, it was easy to understand.
- The results seem to be clearly better than prior works like GLIGEN.

**Weaknesses:**

- Positioning with respect to some prior work.
    - In the related work section, the paragraphs on layout-to-image generation & dataset synthesis contain some highly related works (like InstanceDiffusion or others listed as the second group of data synthesis approaches). However, these works are only briefly described but not contrasted against the proposed method. So, what's the difference to InstanceDiffusion, for example?
    - Why are there no comparisons to InstanceDiffusion in the experiments, or even simpler data augmentation techniques like "copy-paste"?
- The object distributions are simplified compared to real distributions.
    - Using a Gaussian distribution for a discrete variable seems odd. There will be some probability to sample a negative number and if you clip, you ultimately end up with a different distribution than what was estimated, I guess.
    - It seems that all estimated distributions regarding the bounding boxes are independent. I assume that location and areas are dependent. Are the distributions modeled independently on purpose? As in, is it better for simulation to have independent distributions? Or is this just done for simplicity? And was the impact of this evaluated empirically?
- Missing details on the usefulness of synthetic data
    - For the domain specific domain, it seems that the diffusion models were also tuned only with the 200 selected domain-specific images (according to line 212). I found this information to be crucial in judging whether or not the experiments are valid. I suggest to highlight this aspect already in Section 4.1 - and maybe also discuss why this information is curcial. My thinking here is that if you used more data for tuning the diffusion model than you use to train the detectors, it would be unclear where performance gains come from. It could just come from the additional data used to train the diffusion model, rather than the images being synthetic.
    - Also, does the same conclusion on object detector performance gains from Table 2 also hold when scaling up real and synthetic images? From a practical point of view, annotating 800 more images with bounding boxes would likely be affordable in most cases. So, I'm missing an experiment like in Table 2, but with more images for both real and synthetic images, e.g., 1000 and 25000 real and synthetic images.
    - For the general domain, it's great to see in Table 3 that the detector is better when trained on ODGEN data compared to any other synthetic data. However, the real reference points should be (a) the 10k real images and (b) the combination of 10k real and K synthetic images.

**Questions:**

- When you fine-tune the diffusion model with driving data, don't you have issues with limited diversity? I understand that you might only be interested in the same label spaces as defined in the dataset. But for real applications, one is often interested in simulating something that is missing in a dataset. For instance, a driving dataset may contain lots of regular cars, vans and trucks. But you also want to generate emergency vehicles like police cars or ambulances.
- Does the filtering rate of the "corrupted label filtering" step correlate with object occlusions? As in, are there more issues (during either the generation or even the classification/filtering) for occluded/overlapping objects?
- An interesting application of this paper could be to improve language-based object detectors like in [A]. This work relies on GLIGEN and could benefit from ODGEN. The difference would be that the general semantic knowledge of the diffusion model would be leveraged.

References:
- [A] Generating Enhanced Negatives for Training Language-Based Object Detectors. Zhao et al. CVPR'24

**Limitations:**

The limitations are adequately discussed in the appendix.

---

> ### Author Rebuttal · Authors · 2024-08-03
>
> Thanks for your valuable comments. We provide detailed responses below to resolve your concerns.
>
> $\textbf{1. Comparison with InstanceDiffusion:}$
>
> We didn't include InstanceDiffusion since it consists of an UniFusion module which at least requires bounding box labels and segmentation masks as training data. However, our approach and other methods included for comparison only need bounding box labels. We add an additional comparison by employing the open-source InstanceDiffusion model to generate images following the same setups as Tab. 3 in our paper (mAP results are provided in the YOLOv5s/YOLOv7 format):
>
> Method | FID($\downarrow$) | mAP@.50($\uparrow$) | mAP@.50:95($\uparrow$)
> --------|--------|--------|--------
> MIGC | 21.82 | 9.54/16.01 | 4.67/8.65
> InstanceDiffusion | 23.29 | 10.00/17.10 | 5.42/10.20
> ODGEN (ours) | $\textbf{16.16}$ | $\textbf{18.90}$/$\textbf{24.40}$ | $\textbf{9.70}$/$\textbf{14.20}$
>
> We also include InstanceDiffusion in the experiments mentioned in the second part of the global response (YOLO models trained on 80k real images v.s. YOLO models trained on 80k real images + 20k synthetic images). Our approach outperforms InstanceDiffusion as shown by the results in Tab. 2 in the PDF file attached to the global response.
>
> Other works in the second group of data synthesis approaches: Earlier work like LayoutDiffusion [64] is not implemented with latent diffusion models and thus is not included for comparison. Works [21, 37] are designed for the semantic segmentation task while our ODGEN is designed for the object detection task.
>
> $\textbf{2. Comparison with copy-paste:}$
>
> The copy-paste method requires segmentation masks to get the cropped foreground objects, which are not provided by the RF7 datasets used in this paper. Besides, our approach only requires bounding box labels, which are easier to annotate than masks.
>
> $\textbf{3. Datasets synthesis pipeline design:}$
>
> We agree that the current method cannot exactly reproduce the foreground object distributions in the training datasets. The number of objects in an image is sampled from a joint distribution across different categories while the positions and sizes of bounding boxes are independent. The current dataset synthesis pipeline is designed for simplicity and serves as a method to compare the fidelity and trainability of different methods. As illustrated in the limitations part (Suppl. A), the dataset synthesis pipeline has not been fully optimized yet, which would be interesting to explore in future work.
>
> $\textbf{4. Details of the training data:}$
>
> The whole training process on domain-specific datasets, including the fine-tuning on both cropped objects and entire images and the training of the object-wise conditioning module, only depends on the 200 images used for the baseline YOLO detectors training. Besides, the foreground/background discriminator used for corrupted label filtering is also trained and evaluated on the 200 images. We will highlight this point in the revised manuscript to make it clearer. Thanks for your valuable advice.
>
> $\textbf{5. Experiments with more training data:}$
>
> See the experiments added in the parts 2 and 3 of the global response.
>
> $\textbf{6. Experiments for the trainability evaluation on COCO:}$
>
> See the experiments of training YOLO with "80k real v.s. 80k real + 20k synthetic" images added in part 2 of the global response. The current experiments of trainability on COCO (Tab. 3 in our paper) are designed to evaluate the trainability of using synthetic data only. We train YOLO models on synthetic datasets and evaluate them on real COCO data (sampled from the COCO validation set). We find that results of different methods have high degrees of discrimination and our approach achieves improvements by a large margin compared with existing methods.
>
> $\textbf{7. Generalization to novel categories:}$
>
> It is an interesting topic to evaluate the generalization to novel categories not included in the training process. We use ODGEN trained on COCO to synthesize samples containing objects not included in COCO like moon, ambulance, and tiger et al. We show visualized samples in Fig. 4 in the PDF file enclosed in the global response. It shows that ODGEN trained on COCO is capable of controlling the layout of novel categories. However, we find that the generation quality is not stable enough, which may be impacted by the fine-tuning process on the COCO dataset. We are also trying to train a generic ODGEN on about 3000k images covering more than 3000 categories. We hope to get a powerful and generic model covering most common categories in future work.
>
> $\textbf{8. Corrupted label filtering:}$
>
> The label filtering step is not applied to experiments on COCO. The filtering rate of the "corrupted label filtering" step is fixed for experiments on RF7. The proposed discriminator is only designed to justify foreground and background and doesn't discriminate specific categories. It is designed to filter the boxes in which no objects are synthesized. Even if object A is partially occluded by object B, the discriminator is accessible to the information of object B and still judges it as foreground. Therefore, we don't find apparent performance drops on occluded objects. The accuracy of discriminators on the test sets sampled from the RF7 datasets is over 99% (details are provided in Suppl. D.2, Line 505-509).
>
> $\textbf{9. Benefiting other works with our ODGEN}$
>
> It's good to know that work [A] may be benefited from this paper. We also hope that ODGEN can help and inspire the community in the future.
>
> [21]  Y. Jia, et al. Dginstyle: Domain generalizable semantic segmentation with image diffusion models and stylized semantic control. CVPR 2024 workshop.
>
> [37] D. Peng, et al. Diffusion-based image translation with label guidance for domain adaptive semantic segmentation. ICCV 2023.
>
> [64] G. Zheng, et al. Layoutdiffusion: Controllable diffusion model for layout-to-image generation. CVPR 2023.

---

> > ### Author Response · Authors · 2024-08-07
> > **Supplemental: synthesizing overlapping objects**
> >
> > Compared with other cases, cases with overlapping objects are more complex and challenging to synthesize. As shown by the visualized samples provided in Fig.5, 10, 11, and 12 in our paper, our approach has made great progress in generating overlapping objects and outperforms prior works on layout-image consistency.

---

> > > ### Comment · Reviewer_2vbs · 2024-08-11
> > >
> > > I confirm that I read the rebuttal. I appreciate the author's detailed response. I have no further questions to the authors at this point.

---

> > > > ### Author Response · Authors · 2024-08-12
> > > >
> > > > Dear Reviewer 2vbs,
> > > >
> > > > We are glad that all your concerns have been solved. We appreciate your efforts in reviewing our paper and providing feedback during the discussion period.
> > > >
> > > > Authors

---

### Official Review · Reviewer_PYXX · 2024-07-13

**Soundness:** 3
**Presentation:** 3
**Contribution:** 3
**Rating:** 5
**Confidence:** 4

**Summary:**

ODGEN uses a diffusion-based generation model to create novel images to train object detectors. Object bounding boxes along with the object's textual description are given as a conditioning for the generation step. With these generated images they can improve the detector's performance.

**Strengths:**

The proposed generation pipeline can be used to enrich the training dataset with more high-quality images. Since multiple objects can exist in the same image, using bounding boxes along with the textual description seems to be the better approach.

The paper is well written and the evaluation is done with standard datasets. The improvement over the baselines shows the efficacy of the approach.

**Weaknesses:**

1. There is a high generation cost when there are multiple objects in the same image, which reduces the practicality of the proposed approach.
2. It would be interesting to see if the generated images improve the detector's performance where it originally failed due to lack of data, such as when objects are partially occluded.
3. How are the statistics of the object layout taken?
4. Why only 5k images were generated? Can we see some study of gradually adding synthetic data, like 1k,5k,10k,20k, and some plateauing of the detector performance?
4. It will be only fair to compare with other generation approaches with the same post-generation filtering being applied to them. How does ODGEN compare then?
4. In line 111, it is misleading to say "a new method to fine-tune the diffusion model", as it is nothing more than just fine-tuning.

**Questions:**

Please refer to the weakness section.

**Limitations:**

Please refer to the weakness section.

---

> ### Author Rebuttal · Authors · 2024-08-03
>
> Thanks for your valuable comments. We provide detailed responses below to resolve your concerns.
>
> $\textbf{1. Generation cost:}$
>
> Before training and inference, we can generate an offline library of foreground objects to accelerate the process of building image lists. With the offline library, we can randomly pick images from it to build image lists instead of synthesizing new images of foreground objects every time. In the generation stage (Fig. 2c in our paper), our approach pads both the image and text lists to a fixed length. Therefore, the computational cost for inference with offline libraries doesn't increase with more foreground objects. Other methods like InstanceDiffusion [a] and MIGC [b] need more time for training and inferencing with more objects. Taking the models trained on COCO as an example, to generate an image with 6 bounding boxes on a V100 GPU, ODGEN takes 10 seconds, ControlNet takes 8 seconds, and Instance Diffusion takes 30 seconds.
>
> During the inference, we compare using the same offline image library as training against using a totally different image library. We get very close results as shown in the table below (mAP metrics are provided in the YOLOv5s/YOLOv7 format):
>
> Offline Image Library | FID ($\downarrow$) | mAP@.50 ($\uparrow$) | mAP@.50:95 ($\uparrow$)
> -----|-----|-----|-----
> Same as training | 16.01 | 18.90/9.70 | 24.40/14.20
> Different from training | 16.16 | 18.60/9.52 | 24.20/14.10
>
> Both outperform other methods significantly as shown in Tab. 3 of our paper. It indicates that ODGEN is capable of extracting category information from synthesized samples of foreground objects in image lists instead of depending on certain images of foreground objects. ODGEN is not constrained by the image library of foreground objects used in training and can be generalized to other newly generated offline image libraries consisting of novel synthesized samples of foreground objects, which ensures that the use of offline libraries won't reduce the practicality of ODGEN.
>
> $\textbf{2. Detector performance improvement with synthetic data:}$
>
> We provide several visualized detection results of YOLO models trained with and without synthetic data in Fig. 3 of the PDF file enclosed in the global response. It shows that the synthetic data helps detectors detect some occluded objects.
>
> $\textbf{3. Object layout statistics:}$
>
> As illustrated in Sec. 3.3 (Line 167-173) in our paper, the statistics of the object layout are taken from the training dataset. For example, ODGEN trained on 200 images takes the statistics from the 200 images.
>
> $\textbf{4. The number of synthetic images:}$
>
> We have added the ablations of the number of training samples (1k, 3k, 5k, and 10k) in Tab. 1 in the PDF file attached to the global response. It shows that using 5k synthetic images gets results close to using 10k synthetic images and outperforms using 1k or 3k synthetic images under our experimental setups.
>
> $\textbf{5. Post-generation filtering step:}$
>
> See the analysis and experiments provided in part 1 of the global response.
>
> $\textbf{6. Expression:}$
>
> We will change the expression to "We propose to fine-tune the diffusion model with domain-specific images and cropped foreground objects" in the revised manuscript to clarify the difference between our approach and the fine-tuning method on entire images only.
>
> [a] InstanceDiffusion: Instance-level Control for Image Generation, CVPR 2024
>
> [b] Migc: Multi-instance generation controller for text-to-image synthesis, CVPR 2024

---

> ### Author Response · Authors · 2024-08-13
>
> Dear Reviewer PYXX,
>
> We are writing to kindly remind you that the time left for discussion is only about one day. Could you please confirm that you have read the rebuttal and check if you still have any concern about our work? Thanks for your efforts in reviewing this paper and proposing valuable comments.
>
> Authors.

---

> > ### Comment · Reviewer_PYXX · 2024-08-13
> >
> > Thanks for the rebuttal; After reading them I don't have any further questions.

---

> > > ### Author Response · Authors · 2024-08-13
> > >
> > > Dear Reviewer PYXX,
> > >
> > > We are glad that all your concerns have been solved. We appreciate your efforts in reviewing our paper and providing feedback during the discussion period.
> > >
> > > Authors

---

### Official Review · Reviewer_N8et · 2024-07-14

**Soundness:** 3
**Presentation:** 3
**Contribution:** 3
**Rating:** 6
**Confidence:** 4

**Summary:**

This paper proposes the ODGEN method to generate high-quality images conditioned on bounding boxes. ODGEN fine-tunes Stable Diffusion (SD) on domain-specific datasets to enhance image quality in specialist domains, designing a novel strategy to control SD with object-wise text prompts and synthetic visual conditions to alleviate 'concept bleeding'. Experimental results on the COCO-2014 dataset demonstrate that ODGEN surpasses other methods in control capability. Additionally, the authors design a dataset synthesis pipeline using ODGEN, showing that using additional generated training data can improve the performance of state-of-the-art (SOTA) object detectors.

**Strengths:**

+ The ODGEN method proposed in this paper outperforms previous methods and shows significant improvements.
+ This paper is well organized and written. The overall paper is easy to follow.

**Weaknesses:**

There are some unclear implementation details, and a few experiments are missing. Please refer the Questions.

**Questions:**

- When fine-tuning the pre-trained diffusion model, the authors use not only the entire images from the dataset but also the crops of foreground objects (i.e., resized to 512 x 512). I'm curious if this approach would help in generating particularly small objects.

- During actual inference, what is the visual relationship between the image list $c_{il}$ (i.e., in Eqn. 2) and the final generated image? The authors should provide some visual examples to illustrate this.

- The authors mention that the object-wise conditioning used in the paper can alleviate "concept bleeding" in multi-instance generation. I believe it is necessary to conduct experiments on the COCO-MIG benchmark [65], which assesses the model's ability to control both positioning and attributes, to elaborate on the attribute binding capability of the ODGEN.

    [65] Migc: Multi-instance generation controller for text-to-image synthesis, CVPR'24

- Compared to GLIGEN and MIGC, ODGEN achieves better results in overlapping cases. Which module's design primarily contributes to this improvement? As this is a major issue in the industry, I hope the authors can emphasize this in the methods section.

- Have the authors considered open-sourcing the code and model? I believe this would greatly benefit the community.

**Limitations:**

- The proposed method requires fine-tuning the entire Stable Diffusion model and training an additional ControlNet. This often requires significant computational resources, making reproduction challenging.
- I hope the authors can open-source the corresponding model and code, as this would greatly benefit the community.

---

> ### Author Rebuttal · Authors · 2024-08-03
>
> Thanks for your valuable comments. We provide detailed responses below to resolve your concerns.
>
> $\textbf{1. Fine-tuning on foreground objects:}$
>
> Both the model fine-tuning and the object-wise conditioning module are designed to enhance foreground object generation but not limited to small objects. These two parts are conducted in sequence and contribute to synthesizing both large and small objects together. Given an object detection dataset, we fine-tune Stable Diffusion with the proposed approach to make it capable of synthesizing all kinds of foreground objects and provide visual prompts for the training of the object-wise conditional module. The image lists in the object-wise conditioning module provide the information of category and position for foreground objects. As shown by the visualized samples and quantitative results, our approach obtains improvement in generating dense and small objects compared with other methods and achieves better layout-image consistency.
>
> $\textbf{2. Visual relationship between image lists and synthesized images:}$
>
> The image list is designed to provide the information of category and localization for the control of foreground object layouts. The synthesized images share the same category with objects in generated images and are pasted to the same position as the objects in generated images to build image lists. The objects in image lists and objects in synthesized images may all be apples or cakes but have different shapes and colors. More visualized samples are provided in Fig. 2 of the PDF file enclosed in the global response.
>
> $\textbf{3. Attribute binding:}$
>
> Our approach is designed to synthesize object detection datasets that focus on the categories of foreground objects. As a result, the attribute binding is not in the scope now. Our work focuses on the concept bleeding problem of categories while MIGC pays additional attention to attributes. Taking the right part of Fig. 6 in our paper as an example, our ODGEN fixes the concept bleeding problem of categories and generates the concepts of motorcycle and vehicle correctly. We consider to address the challenge of attribute binding in future work to obtain ODGEN models with powerful attribute binding capability by training on datasets containing annotations of attributes (e.g., Visual Genome [a]). We will also highlight the difference between ODGEN and MIGC in the revised manuscript.
>
> $\textbf{4. Overlapping objects:}$
>
> The image list in the object-wise conditioning module contributes to the improvements in object occlusion. Our ODGEN introduces guidance for objects with synthesized visual prompts. We employ an image list to paste the visual prompt for each object separately on different empty canvases. As a result, ODGEN has access to the localization of overlapping objects with visual knowledge in the image list. We ablate this design in the ablations part (the middle part of Fig. 6 in our paper) to show that it contributes to the synthesis of overlapping objects. Experiments also show that our method contributes to the improvement in complex scene synthesis compared with prior works. We will emphasize this part in the method section of the revised manuscript.
>
> $\textbf{5. Open-source:}$
>
> We have plans to open-source the code and model weights.
>
> [a] Krishna R, Zhu Y, Groth O, et al. Visual genome: Connecting language and vision using crowdsourced dense image annotations. International journal of computer vision, 2017, 123: 32-73.

---

> ### Author Response · Authors · 2024-08-13
>
> Dear Reviewer N8et,
>
> We are writing to kindly remind you that the time left for discussion is only about one day. Could you please confirm that you have read the rebuttal and check if you still have any concern about our work? Thanks for your efforts in reviewing this paper and proposing valuable comments.
>
> Authors.

---

> > ### Comment · Reviewer_N8et · 2024-08-13
> >
> > Thanks for your efforts in rebuttal. The response has addressed my concerns.
> > After reading other reviewers' comments and the response, I would like to raise my rating.

---

> > > ### Author Response · Authors · 2024-08-13
> > >
> > > Dear Reviewer N8et,
> > >
> > > We are glad that all your concerns have been solved. We appreciate your efforts in reviewing our paper and providing feedback during the discussion period.
> > >
> > > Authors

---

### Official Review · Reviewer_Fyyh · 2024-07-31

**Soundness:** 3
**Presentation:** 3
**Contribution:** 3
**Rating:** 6
**Confidence:** 3

**Summary:**

It proposes a novel data synthesis pipeline, ODGEN, which uses diffusion models to generate high-quality and controllable datasets for object detection.
They first fine-tune the pre-trained diffusion models on both cropped foreground objects and entire images. Next, they control the diffusion model using synthesized visual prompts with spatial constraints and object-wise textual descriptions.

For each image, the object number per category, bounding box location and size are sampled from estimated normal distributions. The sampled values are used to generate text lists, image lists, and global text prompts which are fed to ODGEN to generate the image.

**Strengths:**

The method addresses a practical problem in machine learning, and new effective techniques are employed in the application of diffusion models to achieve significant better synthesis quality than existing approaches.

For example, two-step encoding (CLIP text tokenizer -> stacking -> encoder) of the textual condition enables the ControlNet to capture the information of each object with separate encoding and alleviates the concept bleeding problem of multiple categories. Also, encoding individual image lists rather than pasting all objects on a single image can effectively avoid the influence of object occlusions.

The experimental validation is thorough, covering multiple benchmarks and providing detailed comparisons with baseline methods.

Ablation analysis is insightful.

**Weaknesses:**

As mentioned in section D2, the encoder architecture (used for image and text embedding) differs for different datasets, which potentially limits the generalizability of the approach. This should be discussed.

For the data scarcity experiment, authors only sample 200 images as the training set for all datasets, which is interesting but not sufficient for the analysis. More extensive evaluation can be done with higher ratios of real:synthetic data for datasets like ApexGame, Robomaster, and Underwater (and even larger datasets like COCO) .

Restricting real images to 200 might lead to sub-optimal “Domain-specific Diffusion Model Fine-tuning”, which maybe the reason why “synthetic only” version in Table 7 doesn’t perform very well. Kindly comment on this.

ODGEN’s and RICO have similar FID score on COCO (while the mAP is quite different), which could be discussed/explained in the paper.

There are no comparisons provided with the fully supervised approach (models trained on complete real training set) on representative datasets. This would help evaluate the trainability of the proposed approach over data augmentation techniques such as copy-paste.

The computational complexity (and training time) of the proposed approach should be discussed and compared with existing methods.

**Questions:**

As mentioned in section D2, the encoder architecture (used for image and text embedding) differs for different datasets, which potentially limits the generalizability of the approach. This should be discussed.

Restricting real images to 200 might lead to sub-optimal “Domain-specific Diffusion Model Fine-tuning”, which maybe the reason why “synthetic only” version in Table 7 doesn’t perform very well. Kindly comment on this.

ODGEN’s and RICO have similar FID score on COCO (while the mAP is quite different), which could be discussed/explained in the paper.

The computational complexity (and training time) of the proposed approach should be discussed and compared with existing methods.

**Limitations:**

Yes.

---

> ### Author Rebuttal · Authors · 2024-08-03
>
> Thanks for your valuable comments. We provide detailed responses below to resolve your concerns.
>
> $\textbf{1. Encoder architectures:}$
>
> In this paper, we change the channel number according to the maximum object numbers that can be found in a single image. For datasets like MRI in which most images contain only one object, we can use fewer channels to make the model more lightweight. For a generic model trained on large-scale datasets, we can set the channel number with a higher enough value that can cover most circumstances. For example, we are working on a model trained on about 3000k images and set the input channels as 150 to make it applicable to images containing at most 50 foreground objects.
>
> $\textbf{2. More extensive evaluation with larger-scale datasets:}$
>
> See the experiments provided in parts 2 and 3 in the global response.
>
> $\textbf{3. ``Synthetic only" not performing well:}$
>
> To the best of our knowledge, the current synthetic data are mostly used to serve as augmentation data instead of training detectors on them only. A similar conclusion was drawn by prior work like GeoDiffusion [a]. Although our approach has achieved improvement in complex scene synthesis, there still exists noticeable gaps between real data and synthetic data. For example, we train ODGEN on 80k images from COCO and synthesize 10k images to train YOLO models from scratch. However, compared with the YOLO trained on 10k real images from scratch, the YOLO model trained on synthetic data only still falls behind (YOLOv7 mAP@.50: trained on 10k synthetic images only 24.40 v.s. trained on 10k real images only 53.20, same labels applied). We look forward to further improvement in future work with more powerful generative models and control methods to narrow the gaps. Besides, as illustrated in the limitations part (Suppl. A), the current dataset synthesis pipeline (Fig. 3 in our paper) is not fully optimized and cannot reproduce the distributions of foreground objects completely, which may also lead to worse performance of training on synthetic data only on RF7 datasets (labels used for COCO experiments are directly obtained from real images).
>
> $\textbf{4. FID and mAP:}$
>
> FID is used to evaluate the distance of distributions between generated samples and real samples by extracting features from images and computing the mean and covariance values. However, it's hard for FID to reflect the condition(layout)-image consistency, which is required by the object detection task and can be reflected by mAP results. Taking Fig. 5 in our paper as an example, ReCo failed to generate the bed within the orange box or generate cups within the purple boxes. Similar misalignment phenomena can be found in Fig. 10, 11, and 12 as well. It will hurt the detector's training and make ReCo perform much worse on trainability (in terms of mAP) than our ODGEN.
>
> $\textbf{5. Comparison with the fully supervised approach:}$
>
> We presented fully supervised experiments with training on real data only as the baseline on RF7 datasets in Tab. 2 of our paper. Similar experiments on the COCO dataset are added to part 2 of the global response. The copy-paste method requires segmentation masks to get the cropped foreground objects, which are not provided by the RF7 datasets employed in this paper. Besides, our ODGEN can be implemented without segmentation masks.
>
> $\textbf{6. Computational complexity and training time:}$
>
> Taking the model for the COCO dataset as an example, our ODGEN shares a very close scale of parameters with ControlNet (Trainable parameters: ODGEN 1231M v.s. ControlNet 1229M. parameters in the UNet of Stable Diffusion are included). We provide the training time for 1 epoch on COCO with 8 V100 GPUs of different methods in the Table below (the training code of MIGC is not open-source and thus not included here):
>
> Method | ReCo | GLIGEN | ControlNet | GeoDiffusion | ODGEN (ours)
> ----------|---------|-------|----------|---------|----------
> Training time (hrs) | 3 | 7 | 4.5 | 3.2 | 5.6
>
> [a] GeoDiffusion: Text-Prompted Geometric Control for Object Detection Data Generation, ICLR 2024.

---

### Author Rebuttal · Authors · 2024-08-03

We thank all reviewers for your efforts in reviewing this paper and providing so many valuable comments. We are glad that all reviewers acknowledge that the performance of our work is better than prior methods on the box-to-image generation task and most reviewers find our paper well-written and easy to follow. Besides, reviewer 2vbs and FcFR point out that the challenge tackled by this paper has many practical use cases, even beyond what this paper lists. We provide detailed responses to resolve reviewers' concerns and will add the discussion and experiments in the rebuttal to the revised manuscript.

We first address several common concerns and provide supplemental figures and tables in the attached PDF:

$\textbf{1. Post-processing with the corrupted label filtering (Reviewer PYXX and FcFR):}$

We design this step to filter some labels when objects are not generated successfully, which may be caused by some unreasonable boxes obtained with the pipeline in Fig. 3 in our paper. For COCO experiments (Tab. 3 in our paper), this step is not applied to any method since we directly use labels from the COCO validation set and hope to synthesize images consistent with the real-world labels. For RF7 experiments (Tab. 2 in our paper), this step is only applied to our ODGEN. For fair comparison, we skip this step on RF7 to compare the generation capability of different methods fairly. We provide results in the Table below:

Table: mAP@.50:.95 ($\\uparrow$) of YOLOv5s/YOLOv7 on RF7. ODGEN (with or without post-processing) leads to greater improvement than other methods on all 7 datasets.
Datasets | Baseline | ReCo | GLIGEN | ControlNet | GeoDiffusion | ODGEN w/o post-processing | ODGEN w/ post-processing
-------------------|-------------------|-------------------|-------------------|-------------------|-------------------|-------------------|------------------
real + synth #|200 + 0|200 + 5000|200 + 5000|200 + 5000|200 + 5000|200 + 5000|200 + 5000
Apex Game|38.3/47.2|25.0/31.5|24.8/32.5|33.8/42.7|29.2/35.8|$\textbf{39.8}$/$\textbf{52.6}$|$\textbf{39.9}/\textbf{52.6}$
Robomaster|27.2/26.5|18.2/27.9|19.1/25.0|24.4/32.9|18.2/22.6|$\textbf{39.0}$/$\textbf{33.3}$|$\textbf{39.6}/\textbf{34.7}$
MRI Image|37.6/27.4|42.7/38.3|32.3/25.9|44.7/37.2| 42.0/38.9|$\textbf{46.1}$/$\textbf{41.5}$|$\textbf{46.1}$/$\textbf{41.5}$
Cotton|16.7/20.5|29.3/ 37.5|28.0/39.0|22.6/35.1|30.2/36.0|$\textbf{40.5}$/$\textbf{42.1}$|$\textbf{42.0}$/$\textbf{43.2}$
Road Traffic|35.3/41.0|22.8/29.3|22.2/29.5|22.1/30.5|17.2/29.4|$\textbf{38.2}$/$\textbf{43.2}$|$\textbf{39.2}$/$\textbf{43.8}$
Aquarium|30.0/29.6|23.8/34.3|24.1/32.2|18.2/25.6|21.6/30.9|$\textbf{32.0}$/$\textbf{38.4}$|$\textbf{32.2}$/$\textbf{38.5}$
Underwater|16.7/19.4|13.7/15.8|14.9/18.5|15.5/17.8|13.8/17.2|$\textbf{18.9}$/$\textbf{21.6}$|$\textbf{19.2}$/$\textbf{22.0}$

It shows that the corrupted label filtering step only contributes a small part of the improvement. Without this step, our method still outperforms the other methods significantly. Results of the other methods were provided in Tab. 6 of our paper as well. In addition, as illustrated in the limitations part (Suppl. A), the current dataset synthesis pipeline is designed to compare different methods and can be improved further in future work. We don't see this part as the main contribution of our paper.

$\textbf{2. Scaling up on COCO (Reviewer Fyyh and 2vbs):}$

We conduct experiments by adding 20k synthetic images to the 80k training images. We train YOLO models on the COCO training set (80k images) as the baseline and on the same 80k real images + 20k synthetic images generated by different methods for comparison. The COCO validation set contains 41k images, we use the labels of 20k images of them as conditions to generate the synthetic set and use the other 21k real images for evaluation. The results are shown in Tab. 2 in the attached PDF file. It shows that ODGEN improves the mAP@.50:95 by 0.5\% and outperforms the other methods.

$\textbf{3. Scaling up on RF7 (Reviewer Fyyh, 2vbs, and FcFR):}$

Apart from the experiments on COCO provided above, we add additional experiments with 1000 training images from the Apex Game and the Underwater Object dataset here. We provide the results of YOLO models trained on the combination of real and synthetic data as below. We conduct experiments on ODGEN, ReCo, and GeoDiffusion. ReCo and GeoDiffusion may benefit from larger-scale training datasets since they need to fine-tune more parameters in both the UNet in Stable Diffusion and the CLIP text encoder. GLIGEN struggles to adapt to new domains and ControlNet performs worse on layout control than ODGEN. So they are not included in this part (also limited by computational resources and time in rebuttal). The corrupted label filtering step is not used for any method.

Table. mAP@.50/mAP@.50:.95 ($\uparrow$) results of ODGEN trained on larger-scale datasets.
Datasets|Apex|Apex|Apex|Underwater|Underwater|Underwater
------|--------|--------|--------|--------|--------|--------
real + synth #|1000 + 0|1000 + 5000|1000 + 10000|1000 + 0|1000 + 5000|1000 + 10000|
YOLOv5s ODGEN (Ours)|83.2/53.5|$\textbf{83.3}$/$\textbf{53.5}$|$\textbf{83.6}$/$\textbf{53.6}$|55.6/29.2|$\textbf{59.6}$/$\textbf{32.5}$|$\textbf{56.3}$/$\textbf{29.8}$
YOLOv5s ReCo|83.2/53.5|78.7/46.9|82.0/46.9|55.6/29.2|55.1/28.4|55.9/29.1
YOLOv5s GeoDiffusion|83.2/53.5|80.0/47.2|82.5/47.5|55.6/29.2|54.2/27.9|54.3/28.0
YOLOv7 ODGEN (Ours)|83.8/55.0|$\textbf{84.4}$/$\textbf{55.2}$|$\textbf{84.0}$/$\textbf{55.0}$|54.6/28.3|$\textbf{58.2}$/$\textbf{29.8}$|$\textbf{62.1}$/$\textbf{31.8}$
YOLOv7 ReCo|83.8/55.0|80.5/50.7|79.2/49.9|54.6/28.3|56.5/28.7|56.4/30.1
YOLOv7 GeoDiffusion|83.8/55.0|81.2/51.0|81.0/50.5|54.6/28.3|57.0/28.9|55.8/28.9

When trained on larger datasets, the baselines (trained on real data only) become stronger but our ODGEN still benefits detectors with synthetic data and outperforms other methods.

---

### Decision · Program_Chairs · 2024-09-25

**Decision:**

Accept (poster)

**Comment:**

The paper receives all positive ratings, where all the reviewers agree that the rebuttal and replies have addressed the major concerns (two reviewers upgraded the rating). Overall, the paper proposes a data generation pipeline via the diffusion model for domain-specific object detection. Comprehensive experiments are provided to validate the effectiveness of the proposed framework. The AC carefully checked the paper, reviews, and rebuttal, and agree with the reviewers about the positive rating. Hence the AC recommends the acceptance decision. It would be highly recommended that the authors include the suggested feedback in the final version (especially the experiments with more real and synthetic data to showcase a broader usage) and release the code to foster the progress in this community.